# Foundation Inference Models for Ordinary Differential Equations

**Johannes R. Hübers** [1 2 *]   **Maximilian Mauel** [3 *]   **David Berghaus** [1 2]   **Patrick Seifner** [1 3]   **Ramsés J. Sánchez** [1 2 3]

## Abstract

Ordinary differential equations (ODEs) are central to scientific modelling, but inferring their vector fields from noisy trajectories remains challenging. Current approaches such as symbolic regression, Gaussian process (GP) regression, and Neural ODEs often require complex training pipelines and substantial machine learning expertise, or they depend strongly on system-specific prior knowledge. We propose `FIM-ODE`, a pretrained Foundation Inference Model that *amortises* low-dimensional ODE inference by predicting the vector field directly from noisy trajectory data *in a single forward pass*. We pretrain `FIM-ODE` on a prior distribution over ODEs with low-degree polynomial vector fields and represent the target field with neural operators. `FIM-ODE` achieves strong zero-shot performance, matching and often improving upon `ODEFormer`, a recent pretrained symbolic baseline, across a range of regimes despite using a *simpler* pretraining prior distribution. Pretraining also provides a strong initialisation for *finetuning*, enabling fast and stable adaptation that outperforms modern neural and GP baselines *without requiring machine learning expertise*. Our pretrained model, code repository, and tutorials are available online[1].

## 1. Introduction

The amortisation of inference procedures, through *pretraining* deep neural networks on large and heterogeneous *synthetic* datasets, is rapidly reshaping AI. This approach underlies many foundation models for time series forecasting (Dooley et al., 2024; Bhethanabhotla et al., 2024; Hemmer & Durstewitz, 2025) and imputation (Seifner et al., 2025b), prior fitted networks for tabular prediction (Müller et al., 2022; Hollmann et al., 2023; 2025), as well as models for causal discovery (Lorch et al., 2022; Kim et al., 2025), mutual information estimation (Gritsai et al., 2025), and dose response prediction in pharmacokinetics (Ojeda et al., 2026b;a).

The idea is simple and (one could argue) stems from simulation-based inference (Cranmer et al., 2020). It consists in shifting the cost of inference from repeated dataset-specific optimization to a single, upfront pretraining phase across diverse synthetic datasets. Such pretraining pushes models to learn *reusable* inference algorithms that (ideally) do not depend on the specific conditioning context. The result is a class of foundation models that enable fast *zero-shot inference* on unseen data.

This trend has recently reached *system identification* in the form of Foundation Inference Models (FIMs). These models infer finite- or infinite-dimensional parametrizations of dynamical systems directly from noisy trajectories *in a single forward pass*. Examples include FIMs for continuous-time Markov chains (Berghaus et al., 2024), stochastic differential equations (Seifner et al., 2025a), and point processes (Berghaus et al., 2026). In this work we focus on ordinary differential equations (ODEs).

ODEs have played a fundamental role in scientific modelling across almost every discipline. They emerged as a language for celestial mechanics (Newton, 1687; Bernoulli, 1712), and remain a default model class for dynamical phenomena. Classical examples include concentration dynamics in molecular reaction networks (Hoff, 1986) and population oscillations in biology (Lotka, 1925; Volterra, 1927). They also provide some of the simplest settings exhibiting chaotic behaviour, with atmospheric convection as a canonical case (Lorenz, 1963). A first step toward amortised ODE inference was made by d'Ascoli et al. (2024) with `ODEFormer`. This model was pretrained on a very large corpus of synthetic ODEs drawn from a *complex* prior distribution, where vector fields are compositions of polynomial, trigonometric, and rational functions. Their goal was to recover the *symbolic expression* of the underlying vector field from noisy ODE solutions. We take `ODEFormer` as our primary baseline, and revisit amortised ODE inference through a simpler lens.

[1]Lamarr Institute For Machine Learning and Artificial Intelligence [2]Fraunhofer IAIS, Germany [3]University of Bonn. Correspondence to: Ramsés J. Sánchez <sanchez@cs.uni-bonn.de>.

*Proceedings of the 43rd International Conference on Machine Learning*, Seoul, South Korea. PMLR 306, 2026. Copyright 2026 by the author(s).

[1]https://fim4science.github.io/OpenFIM/intro.html

Starting from the classical intuition that simple dynamical rules can generate complex patterns (Kadanoff, 1986; 1987; Wolfram, 2002), we ask whether a model pretrained on a much simpler prior can still generalise to real-world systems. Moreover, vector field estimation is only constrained in regions visited by the observed trajectories. This suggests a *local* viewpoint. We therefore ask whether, instead of a symbolic representation that is global in nature, one can represent the vector field locally, and obtain better accuracy in data-rich regimes. With this, our contributions are:

(1) We introduce a pretraining prior distribution over ODEs in dimensions one to three, with polynomial vector fields of degree at most three. We show that a model pretrained on this prior can estimate out-of-distribution vector fields, including vector fields that model human-motion trajectories.

(2) We represent inferred vector fields with neural operators. We show that this *local* representation can match and outperform *global* symbolic representations across a range of settings. In particular, we demonstrate that this (local, neural-operator) representation is *interpretable*, since it can be queried for equilibria, Jacobians, stability types, and phase-space geometry, just as one would query a symbolic vector field.

(3) We show that this pretraining also serves as a strong initialization, enabling *fast and stable finetuning* when the target dynamics are far out of distribution.

## 2. Related Work

To infer an ODE from data is to infer its underlying vector field. Non-parametric vector field inference methods fall mainly into three families, namely symbolic regression, Gaussian process (GP) regression, and neural ODE approaches. The first family aims for *symbolic* vector fields. Genetic programming methods search over symbolic expressions, but typically require time derivative estimates. They therefore rely on clean and densely sampled trajectories (Gaucel et al., 2014; La Cava et al., 2016; Quade et al., 2016; Kronberger et al., 2019; Atkinson et al., 2019; Weilbach et al., 2021), or on complex variational surrogates (Qian et al., 2022). SINDy (Brunton et al., 2016), a prominent alternative (Delahunt & Kutz, 2022; Brunton et al., 2025), assumes that the vector field admits a sparse linear representation in a *predefined* library of basis functions. This makes it sensitive to the choice of library and thus to prior knowledge, in addition to still requiring high-quality derivative information. The second family represents the vector field with Gaussian processes, which makes performance strongly dependent on the choice of the GP prior. Existing approaches have relied on gradient matching (Äijö & Lähdesmäki, 2009), on adjoint-based formulations with

maximum a posteriori estimation (Heinonen et al., 2018) and, more recently, on mean-field variational approximations (Hegde et al., 2022). The third family corresponds to Neural ODEs (Chen et al., 2018; Yildiz et al., 2019; Rubanova et al., 2019; Dandekar et al., 2020), which avoids explicit priors by learning the vector field with neural networks. This flexibility comes at the cost of expensive and unstable training, due to backpropagation through numerical solvers or reliance on slow adjoint methods (Dupont et al., 2019; Finlay et al., 2020; Choromanski et al., 2020; Pal et al., 2021; Zhi et al., 2022). A parallel line of work resorts to neural variational inference, and parametrises both prior and posterior over vector fields with normalizing flows (Xu et al., 2025), but such methods are known to suffer from convergence issues (Adam et al., 2021; Verma et al., 2024). Finally, there is emerging work that leverages LLM agents to model vector fields using programs (Holt et al., 2024).

Overall, these approaches follow *the classical inference paradigm*, in which a model is optimised for a single dataset at a time. They rely on derivative estimation, careful prior design, solver-based training, or delicate variational optimisation, which can lead to complex training pipelines and, therefore, require substantial ML expertise. In contrast, amortised approaches learn an inference procedure once, through pretraining, and then apply it to new systems in a single forward pass. Besides ODEFormer, there are two other attempts at amortised ODE inference, one limited to one dimensional ODEs (Becker et al., 2023), and another that extends ODEFormer to contexts containing more than one trajectory (Şahin et al., 2025).

## 3. Preliminaries

In this section, we briefly introduce the ODE class we focus on, namely *autonomous, first-order* ordinary differential equations, and we formalise the data-driven ODE inference problem.

**Ordinary Differential Equations.** A $d$-dimensional, autonomous, first-order ODE is defined as

$$\frac{d\mathbf{x}(t)}{dt} = \mathbf{f}\left(\mathbf{x}(t)\right), \ t \in \mathbb{R}_+, \ \mathbf{x}(t) \in \mathbb{R}^d. \tag{1}$$

The vector-valued function $\mathbf{f} : \mathbb{R}^d \to \mathbb{R}^d$ is the state-dependent *vector field*, and it fully characterizes the dynamics. Given some initial condition $\mathbf{x}(0)$ in $\mathbb{R}^d$, the ODE solution corresponds to a deterministic trajectory $\mathbf{x}(t)$, which we refer to as the *system state* over time. If the vector field $\mathbf{f}$ is locally Lipschitz in $\mathbf{x}$, the Picard-Lindelöf theorem guarantees that the corresponding initial value problem (IVP) has a unique solution in a neighbourhood of the initial condition (Arnold, 1992). This guarantee forms the foundation for both analytical and numerical approaches to solving ODEs. Appendix A provides additional background on ex-

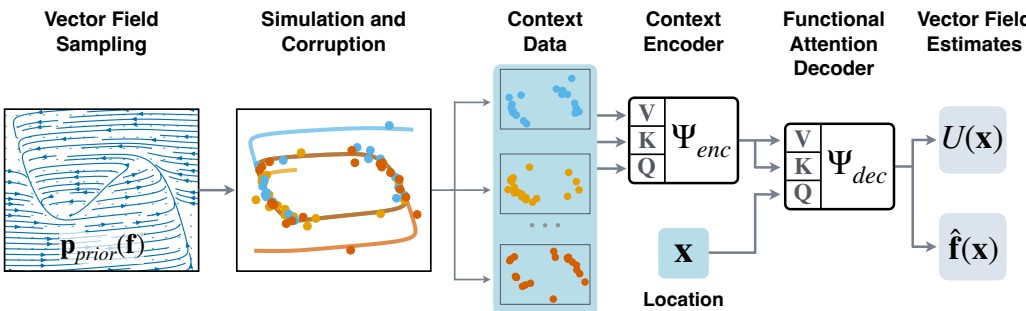

*Figure 1.* Synthetic data generation (left) and `FIM-ODE` architecture (right).

istence and uniqueness results, numerical solution methods, and typical qualitative behaviour.

**ODE Inference Problem.** Consider a dynamical phenomenon whose evolution we observe through noisy and sparse measurements. Let $\mathcal{D}^* = \{(\mathbf{y}_1^*, \tau_1^*), \ldots, (\mathbf{y}_L^*, \tau_L^*)\}$ denote a dataset of $L$ observations recorded at irregular time points $0 \leq \tau_1^* < \cdots < \tau_L^*$. We *assume* that each observation $\mathbf{y}_i^* \in \mathbb{R}^d$ corresponds to a noisy measurement of the (hidden) state $\mathbf{x}(\tau_i)$ of a dynamical system governed by the ODE above (Eq. 1), with an unknown vector field $\mathbf{f}^* : \mathbb{R}^d \to \mathbb{R}^d$. Our goal is to recover $\mathbf{f}^*$ from the observations alone.

More precisely, we seek to learn a parametric function $\hat{\mathbf{f}}_\theta : \mathbb{R}^d \times \mathcal{C} \to \mathbb{R}^d$, with parameters $\theta$ and $\mathcal{C}$ denoting the space of context datasets, such that when conditioning on $\mathcal{D}^*$, $\hat{\mathbf{f}}_\theta$ yields an accurate approximation of $\mathbf{f}^*(\mathbf{x})$ *in the region of state space visited by the observed trajectory*. Finally, we note that data-driven ODE inference is not only NP hard (Cubitt et al., 2012), but also subject to fundamental non-identifiability issues (Miao et al., 2011; Wang et al., 2024b; Casolo et al., 2025). We nevertheless press on. In practice our aim is modest, namely to obtain a useful approximation of the vector field in the regions that are actually supported by trajectory data, and to *generalise* reliably within that regime.

**Notation.** We use $\mathbf{x}(t)$ to denote simulated ODE trajectories, and $\mathbf{x}^*(t)$ to denote hidden ODE processes assumed to underlie observed data. Similarly, $\mathbf{y}(t)$ refers to artificially corrupted trajectories, while $\mathbf{y}^*(t)$ denotes target data. We also distinguish between the simulation discretization step $\Delta t$, and the empirical inter-observation times $\Delta \tau$.

# 4. Foundation Inference Models For ODEs

We now introduce `FIM-ODE`, a pretrained model for *zero-shot* ODE inference from noisy trajectories. Our approach builds upon the Foundation Inference Model (FIM) framework, which amortises dynamical system inference by learning inference procedures during pretraining (Berghaus et al.,

2024; 2026; Seifner et al., 2025b;a). The framework has two components. First, a pretraining prior distribution over the class of dynamical systems of interest. Second, a neural inference model that maps noisy simulated observations back to the parametrization of the underlying dynamics (*e.g.*, the vector field of an ODE). See Figure 1 for an illustration.

## 4.1. Pretraining Prior Distribution over ODEs

Our pretraining distribution is defined by three components, namely a prior over vector fields $p(\mathbf{f})$, a prior over initial conditions $p(\mathbf{x}_0)$, and a corruption mechanism. Accordingly, the data generation pipeline has three stages: (i) sample a vector field $\mathbf{f} \sim p(\mathbf{f})$; (ii) simulate multiple trajectories from each ODE by numerically integrating them from initial conditions $\mathbf{x}_0 \sim p(\mathbf{x}_0)$; and (iii) corrupt the trajectories with noise and subsampling to mimic realistic observations.

**Prior Over Vector Fields**. We consider vector fields whose components are *sparse multivariate low-degree polynomials with random coefficients*, motivated by three considerations. First, many canonical ODEs used to model dynamical phenomena, from the Lorenz system to biological oscillators, are low-degree polynomial systems (see, *e.g.*, the collection in ODEBench (d'Ascoli et al., 2024)). Second, despite their simple algebraic form, polynomial vector fields generate a wide range of behaviors, including fixed points, limit cycles, and chaotic attractors. This aligns with the classical intuition that simple rules generate complex patterns. Third, polynomials are locally Lipschitz, which guarantees existence and uniqueness of trajectories by the Picard–Lindelöf theorem. Each component $f_i : \mathbb{R}^d \to \mathbb{R}$ is constructed as a multivariate polynomial of total degree *at most 3*. We sample coefficients independently from $\mathcal{N}(0, 1)$ and introduce sparsity by randomly masking out both degrees and individual monomials, yielding polynomials with varying structure and interaction patterns. We generate systems in dimensions $d \in \{1, 2, 3\}$. The full generation procedure, including monomial selection and sparsity control, is given in Appendix B.

*Gaussian-process view*. This construction admits a conve-

nient GP interpretation. Conditional on a fixed monomial mask, each $f_i$ is a finite-dimensional Gaussian process. That is, each $f_i$ is a linear combination of deterministic monomial features with Gaussian weights, and its kernel is the inner product of the corresponding feature vectors (equivalently, a sum over the active monomials). Marginalizing over the random mask yields a mixture of such GPs rather than a single GP, which preserves the same basic second-order structure but induces heavier-tailed variability across sampled systems. In either case, the prior is *nonstationary*, meaning that the pointwise variance $\mathrm{Var}(f_i(x))$ increases with $||\mathbf{x}||$ and is dominated by the highest-degree terms (scaling like $||\mathbf{x}||^{2\alpha}$, with $\alpha$ the highest degree, up to direction-dependent constants). See, for example, Figure 6 in the Appendix. Practically, this has consequences for our data generation, which we discuss in our Limitations section (Section 6).

**Trajectory Simulation**. For each sampled vector field, we draw $K$ initial conditions from $\mathcal{N}(0, 1)$, and integrate the ODE forward in time. We define an observation window $[0, 10]$ with 200 equidistant points, so that $\Delta t = 0.05$. We then integrate using Euler's method with 20 steps per observation interval, giving an integration step size of $0.0025$. Finally, we discard systems that produce divergent trajectories, defined as trajectories whose magnitude exceeds $10^2$. This focuses training on bounded regimes, and avoids wasting capacity on numerical blow-ups.

**Trajectory Corruption**. Real measurements are neither perfectly accurate nor uniformly sampled. To expose the model to these conditions, we corrupt simulated trajectories before training. We follow the corruption scheme of ODEFormer (d'Ascoli et al., 2024), that is, we use multiplicative Gaussian noise, which keeps the signal-to-noise ratio in a controlled range, together with random subsampling. Concretely, we perturb the state as $y_i = (1 + \epsilon)x_i$ with $\epsilon \sim \mathcal{N}(0, \sigma^2)$ and $\sigma \in [0, 0.06]$, and we remove observations using an independent Bernoulli mask with probability $\rho \in [0, 0.5]$. All $K$ trajectories from the same system share the same noise scale $\sigma$, but use independently sampled masks. Full details of the corruption procedure appear in Appendix B.

### 4.2. `FIM-ODE`: a Transformer-based Neural Operator Model

We now introduce a model that learns a parametric map $\hat{\mathbf{f}}_\theta : \mathbb{R}^d \times \mathcal{C} \to \mathbb{R}^d$, where $\theta$ are trainable parameters and $\mathcal{C}$ denotes the space of context datasets. Conditioned on a context $\mathcal{D} = \{(\mathbf{y}_{1k}, \tau_{1k}), \dots, (\mathbf{y}_{L_k,k}, \tau_{L_k,k})\}_{k=1}^K$ of $K$ noisy trajectories, the model returns a *local estimate* of the vector field at a query location $\mathbf{x}$. The aim is to approximate the vector field that best explains the observed trajectories, in the region of state-space visited by the data. This requires three capabilities. First, the model must process irregularly

sampled, multi-trajectory observations. Second, it must relate query locations to nearby transitions in state space. And third, it must also generalise across systems with very different temporal and spatial scales. We address these requirements with a neural-operator architecture (Lu et al., 2021) built on attention mechanisms. The design follows an encoder-decoder structure: the encoder embeds the trajectory observations into a permutation-invariant context representation; while the decoder queries this representation at arbitrary spatial locations using cross attention, producing *local* vector field estimates.

**Input Normalization and Scale Invariance.** Different ODEs come with different intrinsic scales. We promote *scale invariance* by normalising each state dimension to zero mean and unit variance, and by re-centring the distribution of inter-observation times $\Delta\tau$ around a target value. The model is trained in this normalised space. Predictions are then mapped back to the original coordinates using the chain rule. Full details appear in Appendix C.

**Transition-based Input Representation.** Rather than feeding raw trajectories, we construct transition features that encode local information. Each consecutive pair $(\mathbf{y}_i, \mathbf{y}_{i+1})$ defines a transition, from which we extract: (i) the current state $\mathbf{y}_i$; (ii) the displacement $\Delta\mathbf{y}_i = \mathbf{y}_{i+1} - \mathbf{y}_i$; (iii) the element-wise squared displacement $\Delta\mathbf{y}_i^2$; and (iv) the inter-observation times $\Delta\tau_i = \tau_{i+1} - \tau_i$. This is motivated by the structure of ODEs: the ratio $\Delta\mathbf{y}_i / \Delta\tau_i$ is a finite-difference estimate of the vector field at $\mathbf{y}_i$, and the squared displacement provides a complementary second-moment feature. Across the $K$ trajectories of lengths $L_1, \dots, L_k$, this extraction yields a set of $J = \sum_{k=1}^K (L_k - 1)$ transition tuples of the form $\tilde{\mathcal{D}} = \{(\mathbf{y}_i, \Delta\mathbf{y}_i, \Delta\mathbf{y}_i^2, \Delta\tau_i)\}_{i=1}^J$.

**Context Encoder.** The encoder maps the context $\tilde{\mathcal{D}}$ into a permutation-invariant representation using self-attention. We first project each feature component independently to dimension $n/4$ using learnable linear projections ($\phi$), then concatenate

$$\mathbf{d}_i = \mathrm{concat}\Big[\phi_{\mathbf{y}}(\mathbf{y}_i), \phi_{\Delta\mathbf{y}}(\Delta\mathbf{y}_i), \phi_{\Delta\mathbf{y}^2}(\Delta\mathbf{y}_i^2), \phi_{\Delta\tau}(\Delta\tau_i)\Big],$$

so that $\mathbf{d}_i \in \mathbb{R}^n$. Next, we apply two layers of linear self-attention (Katharopoulos et al., 2020) to the $n \times J$ matrix $\mathbf{D} = (\mathbf{d}_1, \dots, \mathbf{d}_J)$, yielding a context representation $\mathbf{C} = \Psi_{enc}(\mathbf{D}, \mathbf{D}, \mathbf{D}) \in \mathbb{R}^{n \times J}$.

**Functional Attention Decoder.** Given a query location $\mathbf{x} \in \mathbb{R}^d$, the decoder extracts information from $\mathbf{C}$ via cross attention. We embed the location with a linear map $\phi_{\mathbf{x}}(\mathbf{x}) \in \mathbb{R}^n$, then pass it through $M$ decoder blocks ($\psi$). Each block performs cross attention with queries from the location embedding, and keys and values from $\mathbf{C}$ and returns $\mathbf{h}_i = \psi_i(\mathbf{h}_{i-1}, \mathbf{C}, \mathbf{C}) \in \mathbb{R}^n$, with $\mathbf{h}_0 = \phi_{\mathbf{x}}(\mathbf{x})$. A final MLP maps the result to $\mathbb{R}^d$, so that $\hat{\mathbf{f}}_\theta(\mathbf{x}|\tilde{\mathcal{D}}) = \Psi_{dec}(\mathbf{h}_M(\mathbf{x}))$. This yields our vector field estimator, which

we can evaluate at any point $\mathbf{x}$ in state space, not only at observed states.

This architecture builds on `FIM-SDE` (Seifner et al., 2025a). However, adapting this framework to ODEs requires addressing a different set of identifiability issues. In stochastic differential equations, the stochastic term is typically assumed to have full support, allowing trajectories to explore the state space more broadly, and leading to stronger identifiability guarantees (Bellot et al., 2022; Wang et al., 2024a). By contrast, ODE trajectories constrain the vector field only along the observed paths, leaving its behaviour elsewhere largely underdetermined. This makes the design of the synthetic prior distribution particularly important in `FIM-ODE`, as it encodes which vector-field behaviors are *plausible* away from the observed paths.

**Vector Field Loss with Uncertainty Weighting.** Given a context dataset $\tilde{\mathcal{D}}$, and its associated vector field $\mathbf{f}$, we train `FIM-ODE` by sampling query locations $\mathbf{x}$, and matching predicted vector field values to ground truth ones at $\mathbf{x}$. At each training step, we use a mixed sampling strategy: half of the queries are drawn uniformly over the spatial extent of the observed data, and the other half are drawn from states along the simulated trajectories of the target ODE. The base loss is an MAE between $\hat{\mathbf{f}}_\theta(\mathbf{x}|\tilde{\mathcal{D}})$ and $\mathbf{f}(\mathbf{x})$. A practical issue is that flow magnitudes vary widely across state space. Near the origin, $\|\mathbf{f}(\mathbf{x})\|$ can be close to zero, while other regions may exhibit much larger speeds. Without correction, optimisation overemphasises high-magnitude regions and neglects accuracy near regions with vanishing speed. We therefore follow Seifner et al. (2025a) and use *uncertainty weighting*: an auxiliary head that predicts $U_\theta(\mathbf{x}, \tilde{\mathcal{D}})$, interpreted as a log variance term. The objective becomes

$$\mathcal{L}_\theta = \mathbb{E}_{(\mathbf{x}, \tilde{\mathcal{D}}, \mathbf{f})} \left[ e^{-U_\theta(\mathbf{x}, \tilde{\mathcal{D}})} \|\hat{\mathbf{f}}_\theta(\mathbf{x}|\tilde{\mathcal{D}}) - \mathbf{f}(\mathbf{x})\| + U_\theta(\mathbf{x}, \tilde{\mathcal{D}}) \right],$$

which corresponds to a Laplace likelihood with heteroscedastic scale. The first term down-weights uncertain regions, while the second prevents degenerate solutions. The expectation over $\tilde{\mathcal{D}}$ and $\mathbf{f}$ is taken with respect to the pretraining prior distribution. Query locations $\mathbf{x}$ are sampled using the mixed strategy described above.

# 5. Experiments

This section summarises our experimental setup, including pretraining, datasets, evaluation metrics, and baselines.

**Pretraining**. We pretrain a single 13M parameter model, with 8M parameters for `FIM-ODE`, and 5M parameters for the auxiliary head that models $U_\theta$. Pretraining uses a synthetic dataset of 600K polynomial ODE systems, split into 80K 1D, 210K 2D, and 310K 3D systems. During pretraining, the number of context trajectories per system is sampled uniformly between 1 and 9. For each trajectory, we

sample between 100 and 200 noisy observations. Additional details on data generation, architecture, training hyperparameters, and ablations are provided in Appendices C and G. Finally, let us remark that nothing in our architecture prevents training `FIM-ODE` on higher-dimensional systems. We restrict to three dimensions only because of our current data generation pipeline (see the limitations discussion in Section 6).

**Baselines**. We first compare against `ODEFormer` (d'Ascoli et al., 2024), an 86M-parameter transformer that amortises ODE inference by pretraining on roughly 50M synthetic ODE systems. These are drawn from a complex prior, where vector fields are compositions of polynomial, trigonometric, and rational functions. We then compare against a set of state-of-the-art methods trained under the classical *per-dataset* paradigm. This includes neural approaches such as `GP-DNF` (Xu et al., 2025), Bayesian Neural ODE (`BNeuralODE`) (Dandekar et al., 2020), `NeuralODE` (Chen et al., 2018), `ODE2VAE` (Yildiz et al., 2019), and `LatentSDE` (Solin et al., 2021), as well as GP-based approaches such as `GPODE` (Hegde et al., 2022) and `npODE` (Heinonen et al., 2018). We evaluate `FIM-ODE` across settings that target different challenges, including forecasting, imputation, and inference on real-world human motion trajectories. We compute all trajectories generated from `FIM-ODE` inferred vector fields using `scipy.integrate.solve_ivp`.

## 5.1. Experiment 1: ODEBench

We begin by evaluating the zero-shot inference capability of `FIM-ODE` by comparing it to `ODEFormer`. We use ODEBench, a benchmark introduced with `ODEFormer`, which contains 61 autonomous ODE systems[2] (d'Ascoli et al., 2024). The benchmark includes vector fields composed of trigonometric, exponential, and rational functions, together with reference solutions evaluated on a fixed grid of 512 time points. We follow the experimental protocol of ODEBench and consider two corruption mechanisms (the same of our prior): multiplicative noise $y_i = (1 + \epsilon)x_i$ with $\epsilon \sim \mathcal{N}(0, \sigma^2)$, and random subsampling, where a fraction $\rho$ of observations is removed. The task is to infer the hidden vector field from the corrupted trajectory.

We assess the models on two tasks. The first is *trajectory reconstruction*, which measures whether integrating the inferred vector field from the ground-truth initial condition reproduces the clean reference trajectory on the fixed 512 point grid. The second is *trajectory generalisation*, which evaluates whether the inferred vector field produces accurate trajectories from new initial conditions not present in the context. We measure performance using the variance-

---

[2]We exclude two systems with dimension greater than 3, since `FIM-ODE` is pretrained on systems of at most 3 dimensions.

*Table 1.* ODEBench Trajectory Reconstruction. *Metric*: fraction of systems with variance weighted $R^2 > 0.9$. Higher is better.

| Method | $\rho = 0.0$ | | | $\rho = 0.5$ | | |
|---|---|---|---|---|---|---|
| | $\sigma = 0.0$ | $\sigma = 0.03$ | $\sigma = 0.05$ | $\sigma = 0.0$ | $\sigma = 0.03$ | $\sigma = 0.05$ |
| ODEFormer | 63.1% | 61.5% | 61.5% | 63.9% | 66.4% | 61.5% |
| FIM-ODE | 84.4% | 80.3% | 75.4% | 82.8% | 74.6% | 72.1% |

*Table 2.* ODEBench Trajectory Generalization. *Metric*: fraction of systems with variance weighted $R^2 > 0.9$. Higher is better.

| Method | $\rho = 0.0$ | | | $\rho = 0.5$ | | |
|---|---|---|---|---|---|---|
| | $\sigma = 0.0$ | $\sigma = 0.03$ | $\sigma = 0.05$ | $\sigma = 0.0$ | $\sigma = 0.03$ | $\sigma = 0.05$ |
| ODEFormer | 27.9% | 27.9% | 25.4% | 31.1% | 32.8% | 27.0% |
| FIM-ODE | 29.5% | 32.8% | 29.5% | 27.0% | 27.0% | 28.7% |

weighted $R^2$ score, and report the fraction of test systems achieving $R^2 > 0.9$, following the ODEBench protocol (d'Ascoli et al., 2024). Tables 1 and 2 summarise performance on ODEBench across all corruption settings.

For trajectory reconstruction (Table 1), `FIM-ODE` consistently outperforms `ODEFormer` across all noise and subsampling configurations. For trajectory generalisation (Table 2), the two methods perform comparably, although `FIM-ODE` shows a clearer advantage at less stringent thresholds (Appendix E reports $R^2 > 0.8$ results and mean-squared error scores). These results are notable given that `FIM-ODE` is about *ten times smaller* (8M versus 86M parameters for the vector field predictor) and is pretrained on about *eighty times fewer systems* (0.6M versus 50M), highlighting the benefit of `FIM-ODE` 's local vector field representation.

A key point is that ODEBench contains substantial *out-of-distribution* (OOD) structure relative to our polynomial pretraining prior. Roughly one third of the systems have non-polynomial vector fields, including trigonometric and rational components. Despite this mismatch, Tables 1 and 2 show that `FIM-ODE` reconstructs trajectories accurately and can generalise competitively, already hinting at the value of a local estimator. Table 10 in Appendix E makes this point more explicit by separating *in-distribution* (ID) systems (*i.e.*, polynomial vector fields of degree at most three) from OOD ODEBench systems, showing that the aggregate results in Tables 1 and 2 are *not* driven only by ID cases.

**System Identification Interpretability**. To look more deeply into the structure learned by our local vector-field estimator, we now consider two OOD systems, and one ID system from ODEBench, and perform the same kind of qualitative dynamical-systems analysis that one would usually carry out from a symbolic representation. Given $\hat{\mathbf{f}}_\theta$, we can search for candidate equilibria by minimizing $\|\hat{\mathbf{f}}_\theta\|$, compute the Jacobian matrix, and classify local stability from the spectrum of this matrix. This provides an interpretable diagnostic that is stricter than trajectory reconstruction. Figure 2 shows the corresponding results, and Appendix E.4 provides the details.

We begin with a failure case for `FIM-ODE`: the frictionless pendulum. The ground-truth vector field contains a sine term, which is OOD relative to our prior, and its phase portrait is organised by closed low-energy orbits around the origin. The system has a marginal centre at the downward resting equilibrium, and saddle equilibria at the vertically upright equilibria, for instance at $(\pi, 0)$. `ODEFormer` preserves this conservative structure and recovers a centre near the origin, together with saddle-type equilibria. By contrast, `FIM-ODE` reproduces the observed trajectories but biases the local field near the origin toward weak unstable spirals. This effectively drives nearby states back toward the region covered by the context trajectory, rather than preserving the closed-orbit geometry of the true system. This behaviour is visible in Figure 2(a).

Second, consider the reduced CDIMA model (ODE 42 in ODEBench), whose vector field contains rational terms and is therefore also OOD for `FIM-ODE`. Here, the relevant behaviour is largely controlled by local features in the region visited by the data. The ground-truth system has an unstable spiral equilibrium near $(1.78, 4.17)$. `FIM-ODE` performs strongly in both reconstruction and generalisation, identifies a nearby candidate equilibrium, and correctly classifies it as an unstable spiral (Figure 2(b)). In contrast, `ODEFormer` finds a nearby exact symbolic equilibrium but reverses the local stability, predicting a stable spiral. Thus, despite the rational structure being OOD for `FIM-ODE`, the local vector-field estimate preserves an important qualitative feature of the dynamics that the global symbolic prediction misses.

Finally, consider the Lotka-Volterra (LV) "competition model" (ODE 26 in ODEBench), which is ID for our prior. This system has four equilibria: two stable exclusion states, an unstable extinction state, and a saddle coexistence point whose stable manifold separates the two basins of attraction. `FIM-ODE` partially recovers this structure and fits the observed data well. In particular, it splits the true coexistence saddle at $(1, 1)$ into two nearby saddle-like candidates and recovers a stable node near $(0, 2)$. However, it does not recover the stable node at $(3, 0)$, or the unstable equilibrium at $(0, 0)$, as these regions are only weakly constrained by the available context data (Figure 2(c)). `ODEFormer` also recovers part of the boundary structure, but it does not recover the full coexistence geometry.

Together, these examples illustrate the central trade-off between local and global representations. `FIM-ODE` is only asked to approximate the vector field where the data provide constraints, and local estimates can transfer even when the global functional form is OOD. This local representa-

*(a)* Frictionless Pendulum (ODE 28 in ODEBench)

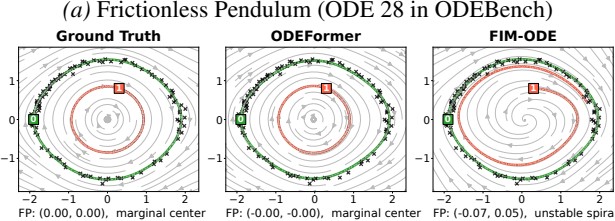

*(b)* Reduced Model for Chlorine Dioxide-iodine-malonic Acid (CDIMA) Reaction (ODE 42 in ODEBench)

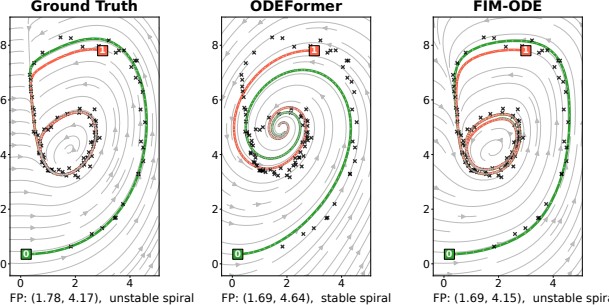

*(c)* Lotka-Volterra: competition model (ODE 26 in ODEBench)

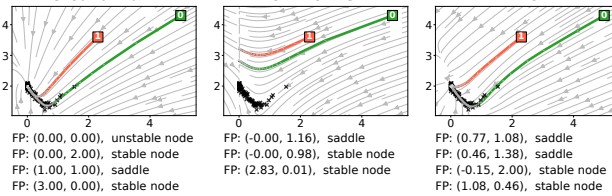

*Figure 2.* Comparison of `ODEFormer` and `FIM-ODE` on three ODEBench systems. Each model infers a vector field from a single corrupted context trajectory, shown with cross markers, obtained by subsampling the ground-truth trajectory (left, green) (with $\rho = 0.5$) and multiplicative noise ($\sigma = 0.03$). The labels $0$ and $1$ indicate the two ODEBench initial conditions. Green and orange curves show the trajectories obtained by integrating the corresponding vector field from these initial conditions. Gray streamlines show the inferred or ground-truth vector fields. We also report the inferred fixed points (FP) and their stability.

tion is interpretable in a dynamical-systems sense: it can be queried for equilibria, Jacobians, stability types, and phase-space geometry, just as one would query a symbolic ODE. By contrast, `ODEFormer` targets a global symbolic expression. This can be powerful when the correct functional family is within distribution, as in the pendulum, but it can also impose global commitments in regions that are not supported by observations, as in the CDIMA and LV examples. Overall, these results demonstrate strong zero-shot ODE inference under realistic corruption, and show that our neural vector fields can support symbolic-style interpretability analyses. Additional results on synthetic polynomial systems, including OOD generalisation to higher-degree polynomials, are provided in Appendix G.

*Table 3.* Oscillator systems used in Experiment 2.

| Van der Pol (VDP) | FitzHugh Nagumo (FHN) |
| --- | --- |
| $\dot{\mathbf{x}} = \begin{bmatrix} x_2 \\ -x_1 + \frac{1}{2}x_2(1 - x_1^2) \end{bmatrix}$ | $\dot{\mathbf{x}} = \begin{bmatrix} 3\left(x_1 - \frac{x_1^3}{3} + x_2\right) \\ \frac{1}{3}\left(0.2 - 3x_1 - 0.2x_2\right) \end{bmatrix}$ |

## 5.2. Experiment 2: Low-data OOD Regime.

We now study how `FIM-ODE` behaves when context trajectories are very short or sparse, and how it can be adapted to this regime through *finetuning*, which amounts to training it on the observed (context) trajectories, *à la* Neural ODE. We outline the details of this finetuning method in Appendix D. We follow the setup of Hegde et al. (2022) and its later reproduction by Xu et al. (2025), and consider two canonical oscillators: Van der Pol (VDP) and FitzHugh Nagumo (FHN). Both systems lie within our polynomial pretraining prior (See Table 3). In this case, the OOD shift is *not* in the vector-field class itself, but in the observation process: the context windows are much shorter than those used during pretraining, and the corruption is additive rather than multiplicative.

**Forecasting the VDP Oscillator.** We simulate the VDP system from the initial condition $\mathbf{x}(0) = (-1.5, 2.5)$ over $t \in [0, 14]$. We use the first half of the trajectory, $t \in [0, 7]$, as context and forecast the second half, $t \in [7, 14]$. From the context window we sample 50 observations corrupted by additive Gaussian noise with variance $\sigma^2 = 0.05$. We also sample 50 target, clean observations on the forecast window for evaluation. We consider two forecasting scenarios: *Task 1* uses uniformly spaced observation times, while *Task 2* uses irregular times drawn uniformly at random. We compare `FIM-ODE` to `ODEFormer` and to the classical baselines listed above, which are trained from scratch on each dataset. We report our results in Table 4.

In this low-data regime, both pretrained models are typically suboptimal relative to classical methods. However, we find that performance depends strongly on the particular noise realisation. The first `FIM-ODE` row in Table 4 reports the result on the exact dataset used by Hegde et al. (2022). To quantify noise sensitivity, we repeated the experiment over 100 independent noise realisations and report the mean and standard deviation as `FIM-ODE` (*Averaged*). The large standard deviations show that, with only a short and noisy context trajectory, the observations do not constrain the underlying dynamics sufficiently for reliable zero-shot forecasting. To further support this point, we augmented the original context dataset with additional noisy trajectories whose initial conditions were sampled from a normal distribution with variance $0.1$ around the original VDP initial condition. All additional trajectories were observed on the same context interval, $t \in [0, 7]$, and on the same time grid

as the original context data. We considered adding 2, 8, and 49 such context trajectories, repeated each setting over 100 noise realisations, and report the largest-context setting as `FIM-ODE` (*Large context*). The remaining context-size results are given in Appendix F. The reduction in standard deviation shows that (indeed) richer context substantially stabilises zero-shot inference.

Finally, we finetune `FIM-ODE` on the fixed dataset of Hegde et al. (2022) by minimising a mean absolute error between model predictions and the context data (see Appendix D for details). This rapidly adapts the model to the low-data regime and substantially improves forecasting performance.

**Imputation in the FHN Oscillator.** We simulate the FHN system from the initial condition $\mathbf{x}(0) = (-1, -1)$ over $t \in [0, 5]$, and sample 50 regularly-spaced observations corrupted by additive Gaussian noise ($\sigma^2 = 0.025$), again following Hegde et al. (2022). To create a structured missing-data regime, we remove all observations whose states fall in the quadrant $x_1 > 0$ and $x_2 < 0$. We then evaluate how well the inferred dynamics recover trajectories that traverse this unobserved region. Table 4 contains our results.

As in the VDP forecasting experiments, the pretrained models are suboptimal, and the zero-shot performance of `FIM-ODE` is sensitive to the particular noise realisation. Adding more context improves the stability of inference, as reflected by the lower variance of `FIM-ODE` (*Large context*). As before, finetuning `FIM-ODE` on the context data yields a sizeable performance gain.

Taken together, the experiments in this section expose a limitation of purely zero-shot inference in low-data regimes. With short context windows and moderate observation noise, the available data may not constrain the dynamics sufficiently for `FIM-ODE` to identify a reliable vector field without adaptation. This uncertainty manifests as sensitivity to the noise realisation. At the same time, the finetuning results show that the pretrained model provides a useful initialization: once dataset-specific information is injected through the context loss, `FIM-ODE` adapts rapidly and achieves strong forecasting and imputation performance.

### 5.3. Experiment 3: Human Motion Capture

Next, we evaluate `FIM-ODE` on the real-world trajectories from the CMU Motion Capture database (Carnegie Mellon University, 2003), using walking and running sequences from subjects 09, 35, and 39. Each recording is a multivariate time series with 50 marker-based coordinates. Following the preprocessing protocol of Hegde et al. (2022) and Xu et al. (2025), we apply principal component analysis (PCA) and project each sequence onto a five-dimensional latent representation, while reporting prediction error in the original observation space via the inverse map. We assess both short-

*Table 4.* MSE on low-data OOD evaluations: VDP forecasting and FHN imputation. Lower is better. T1 and T2 denote Task 1 and Task 2. `ODEFormer` and `FIM-ODE` are evaluated zero shot. `FIM-ODE` (*Averaged*) reports the mean over 100 independent noise realisations, with std shown in parentheses. `FIM-ODE` (*Large context*) uses the largest augmented-context setting. Again, parentheses denote std over 100 trials. `FIM-ODE` (*Finetuned 1 & 2*) are obtained by finetuning `FIM-ODE` on the train sets from Hegde et al. (2022), as explained in Appendix D.

| Method | VDP (T1) | VDP (T2) | FHN |
|---|---|---|---|
| BNeuralODE | 1.45 | 1.68 | 0.24 |
| NeuralODE | 0.29 | 0.55 | 0.18 |
| npODE | 0.16 | 2.08 | 0.08 |
| GPODE | 0.13 | 0.21 | 0.07 |
| ODE2VAE | 0.13 | 0.19 | 0.07 |
| LatentSDE | 0.10 | 0.15 | 0.05 |
| GP-DNF | 0.03 | 0.04 | 0.04 |
| ODEFormer | 0.25 | 0.60 | 0.29 |
| FIM-ODE | 0.90 | 0.43 | 0.40 |
| FIM-ODE (*Averaged*) | 0.8(7) | 2(1) | 0.9(8) |
| FIM-ODE (*Large context*) | 0.35(8) | 0.9(2) | 0.43(6) |
| FIM-ODE (*Finetuned 1*) | 0.2(1) | 0.4(3) | 0.06(3) |
| FIM-ODE (*Finetuned 2*) | **0.028** | **0.022** | **0.029** |

and long-horizon forecasting on held-out test sequences. Short contexts data contain 50 to 100 observations, while long contexts data contain 100 to 250 observations. Both ranges lie within the distribution of context lengths seen during `FIM-ODE` pretraining.

A direct comparison to `ODEFormer` is not meaningful here, since `ODEFormer` can process only a single context trajectory. `FIM-ODE` can condition on multiple trajectories, but it is pretrained for state dimension at most three. We therefore restrict `FIM-ODE`'s latent representation to the first three principal components, and set the remaining components to zero, so that all methods use the same inverse PCA map back to the original measurement space. Table 5 reports test MSE across subjects and forecast horizons.

The zero-shot results show that `FIM-ODE` can transfer effectively to real-world trajectories: for subject 09, for example, `FIM-ODE` achieves the best short-horizon performance among all methods, and remains competitive on long horizons. The picture is different for subjects 35 and 39. Here, zero-shot `FIM-ODE` produces substantially larger errors. This indicates that these motion patterns are less well aligned with the dynamics represented by our pretraining prior distribution. At this stage, however, the source of the degradation is ambiguous. One possible explanation is that an autonomous ODE in the first three PCA coordinates is not an adequate representation of the projected motion dynamics. Another is that the representation is adequate, but the zero-shot inference map does not select the appropriate vector field for these subjects. The finetuning results distinguish between these possibilities: after adapting `FIM-ODE`

*Table 5.* Test MSE for the CMU Motion Capture task. All errors are computed in the original observation space after mapping latent predictions back with the inverse PCA map. `FIM-ODE` is evaluated using the first three principal components. `FIM-ODE` (*Finetuned*) denotes the same model after adaptation on the available subject-specific context data.

| Method | Subject 09 | | Subject 35 | | Subject 39 | |
|---|---|---|---|---|---|---|
| | short | long | short | long | short | long |
| `BNeuralODE` | 25.50 | 21.32 | 23.09 | 20.86 | 53.34 | 39.66 |
| `NeuralODE` | 27.53 | 33.83 | 36.50 | 23.54 | 115.38 | 53.51 |
| `npODE` | 17.91 | 19.76 | 26.24 | 22.83 | 92.80 | 55.94 |
| `GP-ODE` | 9.11 | 8.38 | 10.11 | 11.66 | 26.72 | 21.17 |
| `ODE2VAE` | 9.05 | 8.14 | 9.25 | 10.08 | 25.25 | 21.06 |
| `LatentSDE` | 7.46 | 6.45 | 7.57 | 7.65 | 21.25 | 18.72 |
| `GP-DNF` | 7.03 | 6.04 | **6.72** | **7.03** | 19.43 | **16.21** |
| `FIM-ODE` (3D) | **6.10** | 7.32 | 655.40 | 48.94 | 295.29 | 243.87 |
| `FIM-ODE` (*Finetuned*) (3D) | 7.55 | **5.35** | 6.92 | 11.73 | **15.56** | 19.89 |

on the available subject-specific data, performance improves dramatically for subjects 35 and 39 (see Appendix D for details). Figure 3 showcases the inferred vector fields and trajectories obtained with `FIM-ODE` *(Finetuned)*, while Figure 11 in Appendix D illustrates the gains from finetuning. This shows that the three-dimensional ODE model class can represent useful predictive dynamics for these subjects.

Overall, these results support the broader picture suggested by the previous experiments. Pretraining provides a strong inference prior, and can work remarkably well without adaptation when the target dynamics are well represented by that prior. When this alignment is weaker, zero-shot inference can fail, but modest finetuning can inject the required subject-specific information, and recover strong performance.

## 6. Conclusions

In this work, we extended the Foundation Inference Model framework to *the amortised inference of ordinary differential equations* (ODEs). Our approach has two components. The first is a pretraining prior distribution over ODEs with *polynomial vector fields of degree at most three*. The second is `FIM-ODE`, a neural operator model pretrained on trajectories drawn from this prior. Empirically, we showed that `FIM-ODE` achieves strong zero-shot performance, matching and often outperforming `ODEFormer`, despite being about ten times smaller and trained on about eighty times fewer systems drawn from a much simpler prior. This supports the benefit of a local estimator over global symbolic expressions. We also showed that `FIM-ODE` can be rapidly *finetuned* to challenging out-of-distribution regimes, including real-world human-motion trajectories.

**Limitations.** *The main limitation* lies in the current prior. Our polynomial construction induces a non-stationary scaling, in which typical vector field magnitudes grow with

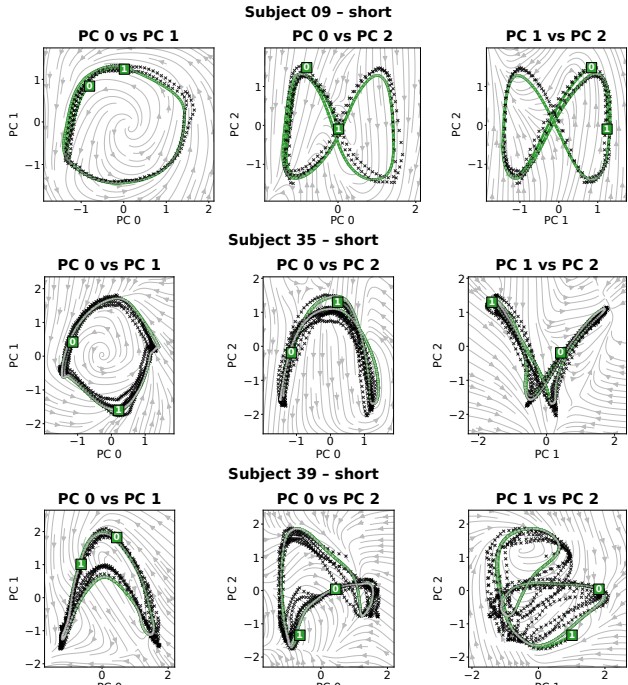

*Figure 3.* Inferred vector fields (gray streamlines) and two trajectories (green) with `FIM-ODE` *(Finetuned)* for short CMU MoCap sequences. Cross markers correspond to target data. Columns show two-dimensional projections of the first three PCA coordinates.

the distance from the origin. As a result, many sampled ODEs generate trajectories that quickly leave the region of interest, or blow up in finite time. This effect becomes more pronounced as dimension increases, creating a *curse of dimensionality* for generating broad, high-dimensional ODE priors that still yield numerically stable and physically plausible trajectories over the desired horizon. This is the main reason we restrict pretraining to dimensions at most three. *A second limitation* is architectural. The maximum state dimension is currently fixed by design, which forces us to choose *a priori* the dimensionality the model can process.

**Future Work.** We plan to address these limitations in two directions. First, we will explore alternative priors, including stationary GP based constructions, with the aim of generating higher-dimensional systems that remain stable while still exhibiting rich dynamics. Second, we will replace the fixed-dimensional design with *axial attention mechanisms* (Ho et al., 2019), so that dimensionality is not hard-coded but inferred from the input. Finally, we will investigate how pretrained `FIM-ODE` weights can serve as a regulariser for equation discovery in high-dimensional settings, following recent progress in this direction (Hinz et al., 2025). We will apply this idea to data-driven discovery benchmarks such as those in Champion et al. (2019) and, more broadly, to dynamical models of evolving text representations (Cvejoski et al., 2023; 2022).

## Impact Statement

`FIM-ODE` lowers the barrier to data-driven ODE inference by enabling scientists to infer dynamical models without training bespoke models, using large datasets, or relying on substantial computational budgets and advanced ML expertise. In this sense, it stands in contrast to standard ML workflows that often require extensive data, compute, and specialization. This may broaden access to interpretable system-identification tools across scientific domains, especially for researchers with limited ML expertise.

A potential risk is that inferred vector fields may be over-interpreted as mechanistic explanations when data are noisy, sparse, and not well represented by the synthetic prior distribution. `FIM-ODE` should therefore be used as a tool for hypothesis generation and model-assisted analysis, together with domain expertise and validation.

## Acknowledgments

This research has been funded by the Federal Ministry of Education and Research of Germany and the state of North-Rhine Westphalia as part of the Lamarr Institute for Machine Learning and Artificial Intelligence.

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

# A. ODE Background

This appendix provides additional theoretical background on ordinary differential equations, including existence and uniqueness theorems, numerical solution methods, and typical dynamical behaviors.

## A.1. Existence and Uniqueness of Solutions

**Peano Theorem — Existence of a Solution.** If the flow field $\mathbf{f}$ is continuous in $\Omega \times [t_0, t_0 + T]$, Peano's theorem states that any adhering initial value problem (IVP) will have at least one solution trajectory. Additionally, the solution trajectories will be contained in an $\epsilon$-ball around $\mathbf{x}_0$ (Teschl, 2012).

**Picard-Lindelöf Theorem — Existence of a Unique Solution.** The Picard-Lindelöf theorem goes one step further and constrains $\mathbf{f}$ to be locally Lipschitz continuous with respect to $\mathbf{x}$ with the Lipschitz constant $L \in \mathbb{R}_+$ independent of $t$ (Teschl, 2012):

$$\|\mathbf{f}(\mathbf{x}_1, t) - \mathbf{f}(\mathbf{x}_2, t)\| \leq L \|\mathbf{x}_1 - \mathbf{x}_2\|. \tag{2}$$

Every IVP that uses an $\mathbf{f}$ upholding those constraints is guaranteed a unique solution trajectory in an $\epsilon$-ball around $\mathbf{x}_0$ and $t \in [t_0, t_0 + T]$.

## A.2. Numerical Solving of Initial Value Problems

In practical environments, analytical methods for solving IVPs typically are not available. Fortunately, IVPs can also be numerically approximated using iterative procedures. One such family of methods, called one-step methods (OSM), is defined as:

$$\mathbf{x}(t + \Delta t) = \mathbf{x}(t) + \int_t^{t+\Delta t} \mathbf{f}(\mathbf{x}(\tau), \tau)\, d\tau \tag{3}$$

or more generally

$$\mathbf{x}(t + \Delta t) = \mathbf{x}(t) + \Delta t \cdot \phi(\mathbf{x}(t), t, \Delta t) \tag{4}$$

where $\Delta t \cdot \phi(\mathbf{x}(t), t, \Delta t)$ can be understood as an approximation of $\int_t^{t+\Delta t} \mathbf{f}(\mathbf{x}(\tau), \tau)\, d\tau$.

Common OSM choices include the Euler method, the trapezoidal method, and the Runge-Kutta method. Euler replaces $\Delta t \cdot \phi$ by $\Delta t \cdot \mathbf{f}(\mathbf{x}(t), t)$, while the Runge-Kutta method needs more evaluations of $\mathbf{f}$ but achieves higher accuracy per step with local accuracy of $\mathcal{O}(\Delta t^4)$ compared to Euler's $\mathcal{O}(\Delta t^1)$.

## A.3. Typical ODE Behaviors

ODEs frequently produce solution trajectories that evolve according to distinctly identifiable structures in state space. Typical ODE features shaping trajectories are:

**Equilibrium Points.** Points $\mathbf{x}_e$ where the system velocity is zero: $\mathbf{f}(\mathbf{x}_e) = 0$. Equilibrium points can be classified as:

- *Attractors (sinks)*: trajectories converge to the equilibrium over time.

- *Sources*: trajectories diverge from the equilibrium.

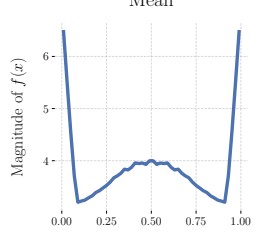 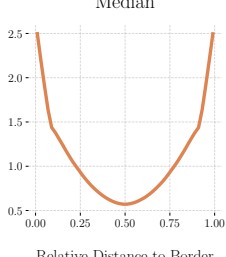 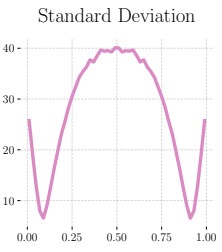

*Figure 4.* Magnitude statistics of vector field points as a function of relative distance to bounding box borders for 1D ODEs. The 1D case exhibits unique behavior due to outlier systems (see Figure 7).

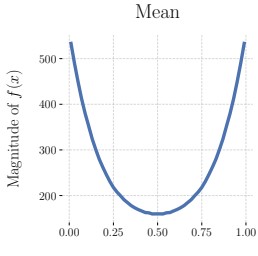 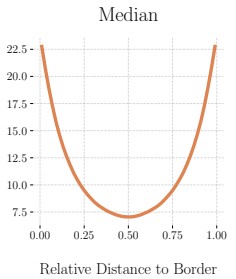 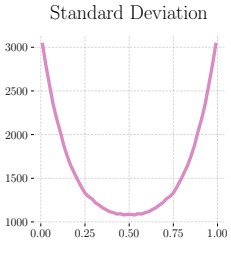

*Figure 5.* Magnitude statistics of vector field points as a function of relative distance to bounding box borders for 2D ODEs.

- *Saddle points*: some directions attract while others repel.

**Limit Cycles.** Trajectories caught in periodic orbits. A limit cycle is characterized by a non-constant trajectory $\mathbf{x}(t)$ that repeats after some time $T > 0$, satisfying $\mathbf{x}(t + T) = \mathbf{x}(t)$.

**Chaotic Behavior.** Chaotic ODEs describe systems whose solution trajectories exhibit extreme sensitivity to initial conditions. This means that two close initial conditions lead to trajectories that diverge rapidly over time, quantified by positive Lyapunov exponents. The Lyapunov exponent $\lambda$ measures how quickly close trajectories diverge: if the separation increases as $\|\delta(t)\| \approx \|\delta(0)\|e^{\lambda t}$, then $\lambda$ is the Lyapunov exponent. A positive Lyapunov exponent ($\lambda > 0$) indicates chaotic behavior.

## B. Data Generation Details and Dataset Statistics

This appendix provides comprehensive details on the synthetic data generation process for training `FIM-ODE`.

### B.1. Polynomial ODE Generation

The generation process targets polynomial ODEs where each component function $f_i$ is a multivariate polynomial of maximal degree $p$:

$$\frac{d}{dt}\begin{bmatrix} x_1 \\ x_2 \\ \vdots \\ x_d \end{bmatrix} = \begin{bmatrix} f_1(x_1, x_2, \ldots, x_d) \\ f_2(x_1, x_2, \ldots, x_d) \\ \vdots \\ f_d(x_1, x_2, \ldots, x_d) \end{bmatrix}, \quad f_i : \mathbb{R}^d \to \mathbb{R} \tag{5}$$

Each component function $f_i$ is defined as an independent multivariate polynomial. Let $\mathbf{X}_d = \{x_1, x_2, \ldots, x_d\}$ be the set of variables. A monomial of degree $p$ is determined by a set $\mathcal{A} = \{\alpha_1, \alpha_2, \ldots, \alpha_d\}$, where $\alpha_i \in \mathbb{N}$ and $\sum_{i=1}^{d} \alpha_i = p$. Each $\mathcal{A}$ defines a monomial term:

$$\mathrm{M}(\mathcal{A}, \mathbf{X}_d) = x_1^{\alpha_1} \cdot x_2^{\alpha_2} \cdots x_d^{\alpha_d} \tag{6}$$

A polynomial is expressed as a weighted sum over all monomials up to degree $p$:

$$f_i(\mathbf{X}_d) = \sum_{j=0}^{p} \sum_{\forall \mathcal{A} \in \boldsymbol{\mathcal{A}_j}} c_{\mathcal{A}} \cdot \mathrm{M}(\mathcal{A}, \mathbf{X}_d), \quad \boldsymbol{\mathcal{A}_j} = \left\{ \mathcal{A} = \{\alpha_1, \ldots, \alpha_d\} \mid \sum_{k=1}^{d} \alpha_k = j \right\} \tag{7}$$

To manage complexity and variability, binary indicator variables $m^{degree} \in \{0, 1\}$ and $m^{monomial} \in \{0, 1\}$ are used to exclude random degrees and monomials:

$$f_i(\mathbf{X}_d) = \sum_{j=0}^{p} m_j^{degree} \cdot \left( \sum_{\forall \mathcal{A} \in \boldsymbol{\mathcal{A}_j}} m_{\mathcal{A}}^{monomial} \cdot c_{\mathcal{A}} \cdot \mathrm{M}(\mathcal{A}, \mathbf{X}_d) \right) \tag{8}$$

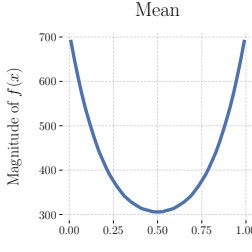 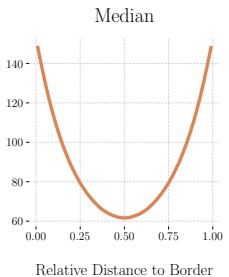 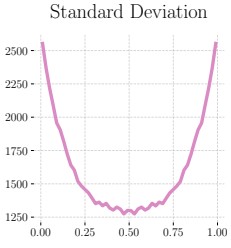

Relative Distance to Border

*Figure 6.* Magnitude statistics of vector field points as a function of relative distance to bounding box borders for 3D ODEs.

After generating a polynomial for each dimension, a global scaling factor $s$ is applied:

$$\frac{d\mathbf{x}(t)}{dt} = s \cdot \mathbf{f}(\mathbf{x}(t)) \tag{9}$$

**Implementation Parameters.** Coefficients $c_{\mathcal{A}}$ are sampled from $\mathcal{N}(0, 1)$. The masks $m^{degree}$ and $m^{monomial}$ are sampled uniformly with constraints ensuring at least one degree and one monomial per polynomial are retained. The scaling factor $s$ is drawn from a uniform distribution over $[0, 2]$.

### B.2. Trajectory Generation and Filtering

Trajectories are generated by solving IVPs from selected initial conditions. For each ODE system, $K = 9$ distinct initial conditions are randomly sampled from $\mathcal{N}(0, 1)$. Trajectories are gathered by numerically integrating the ODE using Euler integration over the time interval $T = [0, 10]$ with $n_{points} = 200$, resulting in a temporal resolution of $\Delta t = 0.05$. The Euler integration uses $n_{intermediate} = 20$ intermediate steps between each point on the grid.

**Data Filtering.** ODEs are discarded if any of their trajectories diverge (any observation exceeds $\delta_{reject} = 10^2$ or becomes non-finite).

### B.3. Dataset Construction and Corruption

The training dataset contains trajectories and matching samples from the vector fields of the ODE systems. To determine where to evaluate the vector field, a bounding box is drawn around the generated trajectories and expanded by a factor $s_{bbox} = 0.2$. Within this expanded bounding box, the ODE's vector field is sampled at $n_{vf} = 10000$ random locations.

**Data Corruption.** Training examples are corrupted using multiplicative Gaussian noise and subsampling. The noise scale $\sigma$ is sampled from $[0, 0.06]$, and the subsampling rate $\rho$ is sampled from $[0, 0.5]$. All trajectories within an ODE system are exposed to the same level of noise, but each trajectory has different points subsampled.

**Dataset Sizes.** The pretraining dataset contains 600,000 polynomial ODE systems spanning dimensionality 1, 2, and 3 (80,000 one-dimensional, 210,000 two-dimensional, and 310,000 three-dimensional systems). A validation set is generated following the same distribution at 10% of the training set size. This dataset size was chosen based on scaling experiments (Appendix G.3). Data characteristics, including spatial biases in vector field sampling, are analyzed in Appendix B.4.

### B.4. Data Characteristics: Vector Field Magnitude Near Boundaries

Understanding the characteristics of the synthetic training data is crucial for interpreting model behavior. This section examines how vector field magnitudes vary spatially within the bounding boxes defined by trajectories.

**Magnitude Distribution by Distance to Boundaries.** During data generation, vector field samples are drawn uniformly within the bounding box defined by the trajectories. This introduces a systematic bias: vector field magnitudes tend to be higher near the boundaries of the bounding box, and variability is also most pronounced near boundaries. Figures 4, 5, and 6 show this pattern across dimensions 1, 2, and 3.

**1D Outlier Behavior.** The 1D case exhibits distinct behavior compared to 2D and 3D. According to the mean statistics, the magnitude is highest at the borders but also increases in the center of the interval. This pattern is caused by outlier ODEs

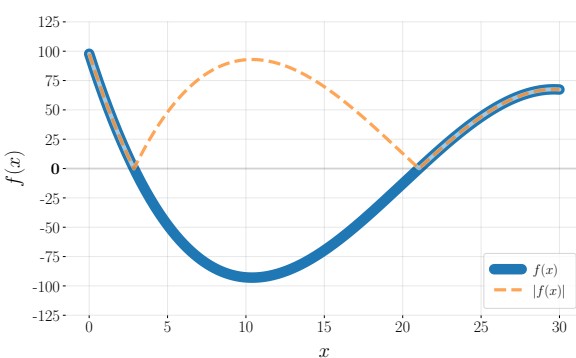

*Figure 7.* Example 1D outlier ODE. Vector field (blue) and magnitude of vector field (orange). The magnitude changes abruptly when the vector field crosses the $x$-axis, creating the distinctive pattern observed in 1D statistics.

where the vector field crosses the $x$-axis, resulting in instantaneous magnitude changes (Figure 7). In contrast, 2D and 3D cases show more consistent patterns without such outlier influence.

**Implications for Training.** As the dimension increases, a larger proportion of uniformly sampled points lie near the boundaries of the bounding box (a consequence of the curse of dimensionality). Meanwhile, the trajectories remain concentrated near the center by construction (since the initial conditions are sampled from $\mathcal{N}(0, I)$). This creates a growing mismatch between where the vector field is sampled for training and where trajectory observations are concentrated. To mitigate this, we employ a mixed sampling strategy: 50% of training locations are drawn from observed trajectory points, and 50% are sampled uniformly within the bounding box.

## C. Architecture and Training Details

This appendix provides comprehensive details on the `FIM-ODE` architecture, normalization schemes, and training procedures.

### C.1. Architecture Specifications

The implemented model has an embedding dimension of $n = 256$. The encoder stack consists of 2 transformer layers, while the decoder stack uses 8 layers. Both the encoder and decoder blocks use multi-head attention with 8 heads and apply the Gaussian Error Linear Units (GELU) activation function. The final MLP consists of three layers with a hidden dimension of 1024.

The model supports systems up to dimension $d \leq 3$ through a padding scheme: when the model receives input with a system of lower dimensionality, the input is padded with zeros in the remaining dimensions.

### C.2. Normalization Schemes

To generalize across systems on different spatial and temporal scales, the model uses two instance normalization schemes.

**Spatial Normalization: Reversible Instance Normalization.** All observations are normalized per dimension using:

$$\text{IN}(\mathbf{y}(t)) = \frac{\mathbf{y}(t) - \mu_\mathbf{Y}}{\sigma_\mathbf{Y}} \tag{10}$$

where $\mu_\mathbf{Y}$ and $\sigma_\mathbf{Y}$ are the mean and standard deviation computed per dimension over the truncated observation set $\mathbf{Y}$ (excluding the last observation of each trajectory due to feature extraction).

The model is trained to predict the normalized flow field:

$$\frac{d\,\text{IN}(\mathbf{y}(t))}{dt} = \hat{\mathbf{f}}_\theta(\text{IN}(\mathbf{y}(t))) \tag{11}$$

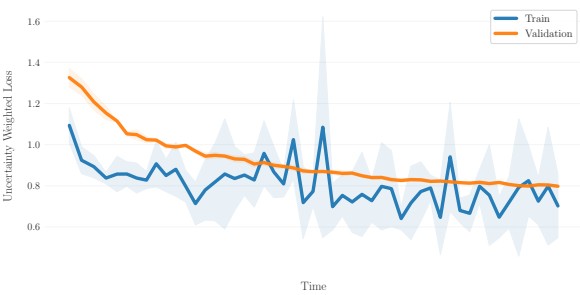

*Figure 8.* Weighted $\mathcal{L}_1$ training and validation losses during pretraining. Loss is weighted according to learned uncertainty estimates.

To return predictions to the original space, the chain rule yields:

$$\frac{d\mathbf{y}(t)}{dt} = \sigma_{\mathbf{Y}} \cdot \hat{\mathbf{f}}_\theta(\text{IN}(\mathbf{y}(t))) \tag{12}$$

Thus, denormalization is implemented by scaling the predicted vector field by $\sigma_{\mathbf{Y}}$.

**Temporal Normalization: Delta Log Centering.** The times are normalized based on $\Delta t$. The $\Delta t$ values are centralized around a target value $\Delta\tau_{target} = 0.01$ while ensuring all $\Delta t > 0$:

$$C(t) = \gamma \cdot t, \quad \gamma = \Delta\tau_{target} \cdot \exp(-\mu_{\log(\Delta t_{\mathbf{Y}})}) \tag{13}$$

where $\mu_{\log(\Delta t_{\mathbf{Y}})}$ is the mean of the logarithms of the time gaps in observation time.

Temporal denormalization also reduces to multiplication by $\gamma$:

$$\frac{d\mathbf{y}(C(t))}{dt} = \hat{\mathbf{f}}_\theta(\mathbf{y}(C(t))) \cdot \gamma \tag{14}$$

### C.3. Training Procedures

Our training procedure extends the `FIM-SDE` methodology (Seifner et al., 2025a) with ODE-specific components.

**Vector Field Training Mode.** The primary training mode evaluates the model at sampled locations and compares predictions against ground truth flow values. During each training iteration, we sample locations $\mathbf{L}$ using a 50-50 split:

$$\mathbf{L} = \mathbf{L}_{traj} \cup \mathbf{L}_{random}, \quad \text{where} \quad |\mathbf{L}_{traj}| = |\mathbf{L}_{random}| \tag{15}$$

where $\mathbf{L}_{traj}$ contains all observed trajectory points (from the $K$ trajectories used as context) and $\mathbf{L}_{random}$ contains an equal number of uniformly sampled points within the bounding box of the trajectories (expanded by 20%). This dual sampling strategy ensures the model learns both trajectory reconstruction and generalization to nearby regions of state space.

**Trajectory / neural ODE Training Mode.** This training method is based just on trajectories and requires no knowledge of the underlying vector field. We use it for finetuning the pretrained model to specific ODEs. It is outlined in Appendix D.

**Normalized Training.** All loss computations occur in normalized space (after applying reversible instance normalization and delta log centering). This is critical for handling systems with vastly different scales and ensures the uncertainty weighting operates on comparable magnitudes across all systems.

**Training Configuration.** We train using the AdamW optimizer (Loshchilov & Hutter, 2019) with weight decay $1 \times 10^{-4}$, learning rate $1 \times 10^{-5}$, batch size 64, 10% dropout, and gradient clipping (max norm 10). The number of trajectories per batch is randomly varied between 1 and 9 to teach the model to handle variable context sizes. Training was performed over five days on four NVIDIA A40 GPUs (48 GB each), optimizing approximately 13 million parameters (8M for FIM-ODE, 5M for uncertainty estimation).

**Pretraining Dataset.** The pretraining dataset contains 600,000 polynomial ODE systems (80K 1D, 210K 2D, 310K 3D), with a 10% validation split following the same distribution. This is comparable in scale to the `FIM-SDE` pretraining (Seifner et al., 2025a) but adapted for the deterministic setting with multiple trajectories per system. Ablations on dataset size (Appendix G.3) show that performance scales with data but with diminishing returns beyond 100,000 systems for the tested model capacity.

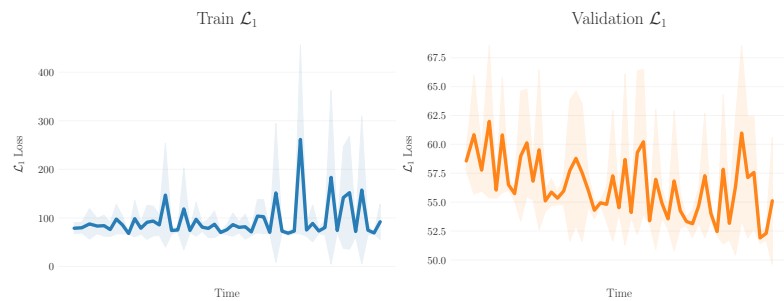

*Figure 9.* Unweighted $\mathcal{L}_1$ training (left) and validation (right) losses during pretraining.

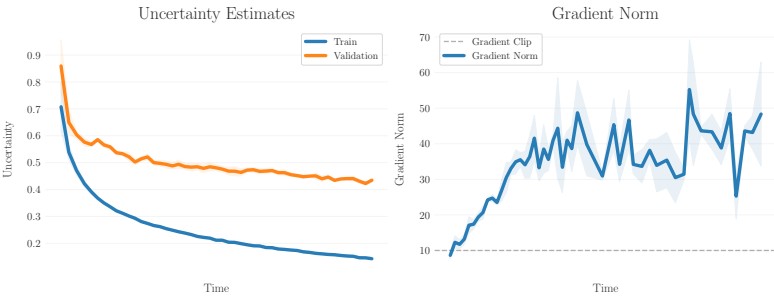

*Figure 10.* Uncertainty estimates for training and validation (left), and gradient norm evolution (right) during pretraining.

### C.4. Training Statistics

This section provides detailed training statistics for the `FIM-ODE` model, including loss curves, uncertainty estimates, and gradient norms during pretraining.

**Loss Evolution.** Figure 8 shows the weighted $\mathcal{L}_1$ training and validation losses throughout pretraining. The training losses are computed using partially randomized vector field locations and random subsets of trajectories, while validation losses are evaluated on all vector field locations conditioned on all available trajectories. This accounts for the smoother appearance of the validation curve. Although the loss curves indicate that further training could yield marginal improvements, the rate of improvement has slowed considerably by the end of training.

**Unweighted Loss Comparison.** Figure 9 presents the unweighted $\mathcal{L}_1$ losses. The training $\mathcal{L}_1$ is consistently higher than the validation $\mathcal{L}_1$, which is explained by the different conditioning: during training, the model accesses only a subset of trajectories, whereas during validation all trajectories are provided. Since vector field predictions empirically improve when conditioned on more input trajectories, the validation loss is correspondingly lower.

**Uncertainty and Gradient Norms.** Figure 10 shows the evolution of uncertainty estimates and gradient norms. The uncertainty estimate, which the model uses to weigh loss terms, decreases during training, indicating increasing confidence in predictions. The gradient norm exhibits an upward trend during training, though gradient clipping (max norm 10) prevents this from destabilizing training. The increasing gradient magnitude in later training stages suggests the model is making finer adjustments to capture subtle features of the flow fields.

## D. Finetuning `FIM-ODE`

We finetune `FIM-ODE` on each benchmark dataset using a context-reconstruction objective. Given an observed context trajectory, the model encodes the data into a vector-field representation $\mathbf{f}_\theta$. During finetuning, this vector field is integrated forward from observed initial states, and the model parameters are updated so that the resulting trajectories reconstruct the context observations.

All finetuning runs optimize the mean absolute error in the model's normalized state space. We use the Adam optimizer and apply gradient clipping with maximum norm 1.0.

**Single-shooting training.** For the experiments in Section 5.2, we use a full-trajectory, or single-shooting, objective. At each

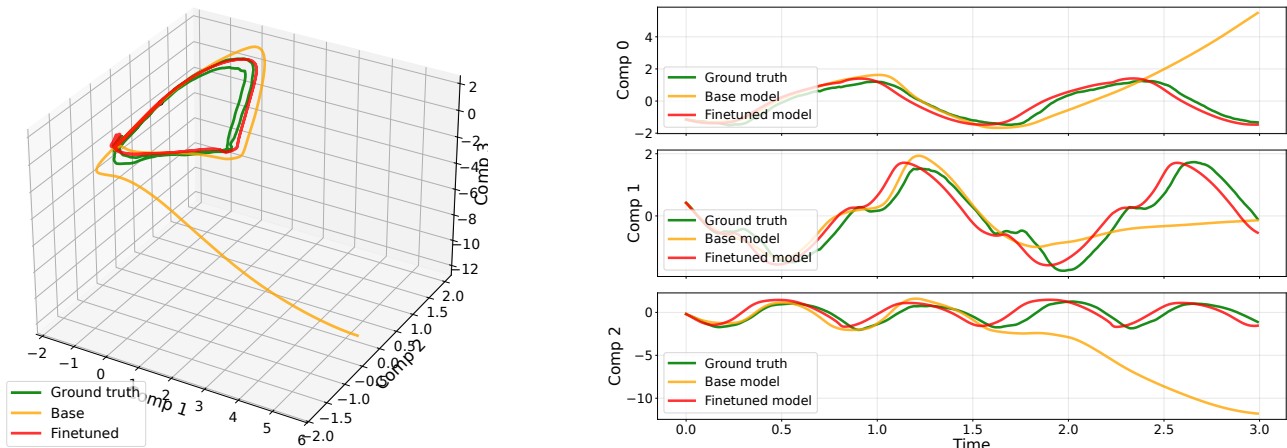

*Figure 11.* Finetuning task: MoCap 35 with short context length

gradient step, the model starts from the first observation $x(t_0)$ and integrates the inferred vector field across all subsequent observation times.

Each interval $[t_i, t_{i+1}]$ is subdivided into $n_{\text{inner}}$ internal solver steps. We use an improved-Euler update of the form

$$\hat{x}(t_{i+1}) = x(t_i) + \Delta t_i \cdot f_\theta\big(x(t_i) + \tfrac{\Delta t_i}{2} f_\theta(x(t_i))\big), \tag{16}$$

where $\Delta t_i = (t_{i+1} - t_i)/n_{\text{inner}}$. The reconstruction loss is then computed between the predicted and observed states along the full context trajectory.

**Multiple-shooting training.** For the MoCap experiments in Section 5.3, we instead use a multiple-shooting objective. This is better suited to longer trajectories, where integrating from the first observation over the entire sequence can make the optimization unstable and overly sensitive to errors accumulated early in the rollout.

Let $(x_i, t_i)_{i=1,\ldots,\ell}$ denote the observed discrete trajectory. The full trajectory is used as context to infer a single ODE vector field. This vector field is then integrated from several initial conditions selected along the observed time grid. Each initial condition defines a short predicted rollout $(\hat{x}_i)_{i=0,\ldots,n_{\text{steps}}}$, synchronized with the corresponding segment of the observed trajectory.

In contrast to the single-shooting experiments, where we use improved-Euler updates, the multiple-shooting rollouts are integrated using fourth-order Runge-Kutta updates. For each rollout, we compute the mean absolute reconstruction error

$$\frac{1}{n_{\text{steps}}} \sum_{i=1}^{n_{\text{steps}}} \|\hat{x}_i - x_i\|_1 \,.$$

The final training loss is obtained by summing these reconstruction errors over all selected initial conditions.

In all multiple-shooting experiments, the initial conditions are chosen regularly along the time grid, starting from the first observed point. We use $n_{\text{steps}} = 25$ rollout steps. To ensure that neighbouring shooting intervals overlap by approximately half their length, we choose

$$n_{\text{IC}} = \left\lfloor \frac{2\ell}{n_{\text{steps}}} \right\rfloor$$

equally spaced initial conditions.

| System | *Finetune 1*: Train-loss selection (top-10) mean $\pm$ std | *Finetune 2*: Test-MSE selection best |
|---|---|---|
| VDP T1 | $0.178 \pm 0.117$ | 0.028 |
| VDP T2 | $0.423 \pm 0.328$ | 0.022 |
| FHN | $0.064 \pm 0.033$ | 0.030 |

*Table 6.* Test MSE under two checkpoint selection strategies. *Train-loss selection* reports the mean $\pm$ std of the test MSE at the 10 epochs with lowest training reconstruction loss. *Test-MSE selection* reports the test MSE of the best checkpoint chosen by the held-out evaluation metric.

### D.1. Checkpoint selection.

The checkpoint-selection protocol depends on the available data split. For the VDP and FHN tasks, only a single short training trajectory is available and there is no separate validation trajectory. We therefore monitor the forecasting MSE on the held-out evaluation window every five epochs and retain the checkpoint with the lowest value. We refer to the results obtained with this selection rule as `FIM-ODE` (*Finetuned 2*) in Table 4 of the main text. This uses the evaluation window for checkpoint selection and therefore does *not* constitute a fully independent test protocol. We report this limitation explicitly, since no separate validation data are available in these benchmarks.

To quantify the effect of this choice, Table 6 compares this selection strategy with a purely training-loss-based alternative. In the latter case, we identify the ten checkpoints with the lowest context-reconstruction loss and report the mean and standard deviation of their test-window MSE. We refer to the results obtained with this training-based variant as `FIM-ODE` (*Finetuned 1*) in Table 4. The comparison shows that selecting by training reconstruction loss alone can lead to substantially worse and more variable performance, especially for the VDP systems which deal with forecasting. This indicates that low reconstruction error on a short context trajectory is not always a reliable proxy for extrapolation quality.

### D.2. Van der Pol — Task 1

The training context consists of a single trajectory of $T_{\text{train}} = 50$ uniformly-spaced observations on $[0, 7)$. The test set comprises 50 uniformly-spaced points on the held-out window $[7, 14]$. No validation data is available (see the checkpoint-selection D.1 caveat above).

| Hyperparameter | Value |
|---|---|
| Mode | full trajectory |
| Epochs | 300 |
| Learning rate | $5 \times 10^{-6}$ |
| Weight decay | $10^{-4}$ |
| Integrator | improved Euler |
| $n_{\text{inner}}$ | 5 |
| Best epoch | 90 |
| Test MSE | 0.02836 |
| Time per epoch | $\approx 4.8\,\text{s}$ |
| Time to best model | $\approx 7\,\text{min}$ |
| Total runtime | $\approx 24\,\text{min}$ |

*Table 7.* Finetuning configuration and timings for VDP Task 1 (CPU).

### D.3. Van der Pol — Task 2

The training context is a single trajectory of $T_{\text{train}} = 50$ observations sampled non-uniformly on $[0, 7)$. The test set consists of 50 points subsampled from the held-out window $[7, 14]$ (fixed random seed). No validation data is available ((see the checkpoint-selection D.1 caveat above).

| Hyperparameter | Value |
|---|---|
| Mode | full trajectory |
| Epochs | 300 |
| Learning rate | $5 \times 10^{-6}$ |
| Weight decay | $10^{-4}$ |
| Integrator | improved Euler |
| $n_{\text{inner}}$ | 5 |
| Freeze encoder | No |
| Best epoch | 55 |
| Test MSE | 0.02230 |
| Time per epoch | $\approx 4.8\,\text{s}$ |
| Time to best model | $\approx 4\,\text{min}$ |
| Total runtime | $\approx 24\,\text{min}$ |

*Table 8.* Finetuning configuration and timings for VDP Task 2(CPU).

## D.4. FitzHugh–Nagumo (FHN)

The FHN training trajectory contains a structured *missing region* (the phase-space quadrant $x_1 > 0$, $x_2 < 0$), leaving $T_{\text{train}} = 38$ of the 50 time points observed. The 12 missing states are withheld entirely from training: neither their values nor their time indices appear in the context or the training loss. The 38 observed points are passed as a single flat sequence in chronological order, so consecutive timestamps across the gap boundary carry a large interval ($\approx 13\times$ the nominal $\Delta t$); the model must integrate across this interval as part of the full-trajectory loss, which motivates the larger $n_{\text{inner}}$ value. The held-out test set consists of the 12 missing time points inside the gap, and test MSE is evaluated there. No validation data is available; see the model-selection caveat above.

| Hyperparameter | Value |
|---|---|
| Mode | full trajectory |
| Epochs | 200 |
| Learning rate | $5 \times 10^{-6}$ |
| Weight decay | $10^{-4}$ |
| Integrator | improved Euler |
| $n_{\text{inner}}$ | 20 |
| Freeze encoder | No |
| Best epoch | 180 |
| Test MSE | 0.02975 |
| Time per epoch | $\approx 13.7\,\text{s}$ |
| Time to best model | $\approx 41\,\text{min}$ |
| Total runtime | $\approx 46\,\text{min}$ |

*Table 9.* Finetuning configuration and timings for FHN (CPU). The higher per-epoch cost relative to VDP reflects the larger number of inner steps needed to accurately integrate across the missing-region gap.

## D.5. Motion Capture (MoCap)

The MoCap tasks' training differs from the Van der Pol and FitzHugh-Nagumo tasks' finetuning in that there are multiple context trajectories, as well as separate validation and testing sequences. We choose the best model based on the neural ODE loss for the validation trajectories.

We found it useful to regularize finetuning by injecting noise at each trajectory step, or also at the initial conditions. Throughout all MoCap finetuning tasks, we employ the same noise level of $\sigma_i = \frac{\Delta t}{5}$ where $\Delta t = t_{i+1} - t_i$ is the time step in ODE solving. For all tasks we trained for either 200 or 800 epochs although in most cases as few as 20 can have very

good results. Figure 11 illustrates the performance gain after finetuning.

Let's remark that finetuning on few trajectories requires little memory and does not benefit much from availability of a GPU, enabling practitioners to improve `FIM-ODE`'s inference capabilities at low cost.

## E. Additional Experiments and Results: ODEBench

### E.1. ID/OOD Split

Tables 10 and 11 provide additional results for the ODEBench evaluation. The main text reports aggregated metrics, which summarize overall performance but do not by themselves reveal the extent to which the models generalize outside the pretraining distribution of `FIM-ODE`. To make this distinction explicit, we split ODEBench into two groups: systems whose vector fields are polynomial with degree at most three, which we regard as in-distribution (ID) for `FIM-ODE`, and all remaining systems, which we regard as out-of-distribution (OOD). This ID/OOD distinction applies to `FIM-ODE`. The corresponding systems are not OOD for `ODEFormer`.

Table 10 reports the percentage of trajectories with $R^2 > 0.9$, separately for trajectory reconstruction and trajectory generalization. For trajectory reconstruction, `FIM-ODE` performs strongly on both ID and OOD systems, especially in one and two dimensions, and consistently improves over `ODEFormer` across most corruption settings. Thus, the strong reconstruction performance reported in the main text is not driven only by ID systems, but also reflects transfer to vector fields outside the polynomial pretraining family.

For trajectory generalization, the picture is more nuanced, as expected. In one dimension, `FIM-ODE` achieves comparable performance on ID and OOD systems. In two dimensions, its performance decreases on OOD systems, consistent with the fact that trajectory generalization requires extrapolating the inferred vector field beyond the region directly constrained by the context trajectory. Even in this setting, however, `FIM-ODE` remains broadly comparable to `ODEFormer`, whose performance also decreases from one to two dimensions despite the different pretraining regime.

### E.2. MSE Metrics

Finally, Table 11 complements the thresholded $R^2$ metric with median MSE and MSE quantiles. We report these results separately by dimension, since MSE values are not directly comparable across systems with different scales. The MSE results are consistent with the thresholded metrics: `FIM-ODE` achieves lower median reconstruction error in all dimensions, and lower median generalization error in two and three dimensions. In one-dimensional generalization, `ODEFormer` has a slightly lower median MSE, while `FIM-ODE` remains competitive. Overall, these results support the interpretation that the local representation learned by `FIM-ODE` is particularly beneficial for reconstruction and can also support meaningful OOD generalization when the relevant region of state space is sufficiently covered by the observed data.

### E.3. $R^2 > 0.8$ **case**

The $R^2$ score (coefficient of determination) measures trajectory reconstruction quality as $R^2 = 1 - \frac{\sum_i (x_i - \hat{x}_i)^2}{\sum_i (x_i - \bar{x})^2}$, where $\hat{x}_i$ are predictions, $x_i$ are ground truth values, and $\bar{x}$ is the mean. For multi-dimensional systems, we use the variance-weighted $R^2$, where each dimension's $R^2$ is weighted by its proportion of total variance across all dimensions.

Table 2 of the main text and Table 12 suggest that the relative performance depends strongly on how strict the success criterion is. At the stringent threshold $R^2 > 0.9$ (Table 2), the two models are broadly comparable, with no consistent winner across noise and subsampling settings. However, when we relax the threshold to $R^2 > 0.8$ (Table 12), `FIM-ODE` shows a clearer edge. Without subsampling ($\rho = 0$), `FIM-ODE` increases the success rate from 35.2% to 41.8% at $\sigma = 0$, from 32.8% to 41.8% at $\sigma = 0.03$, and from 34.4% to 38.5% at $\sigma = 0.05$. Under subsampling ($\rho = 0.5$), the two methods are closer, but `FIM-ODE` remains competitive and often improves robustness to noise, for example 39.3% versus 33.6% at ($\sigma = 0.05$). Overall, these results indicate that `FIM-ODE` more frequently captures the coarse structure of the dynamics well enough to yield useful, though not perfect, long horizon generalisation, which becomes visible once the evaluation allows moderate errors.

*Table 10.* ID/OOD ODEBench results using the thresholded $R^2$ metric. We report the percentage of trajectories with $R^2 > 0.9$ for reconstruction and generalization, separately for `FIM-ODE` and `ODEFormer`. ID systems are those with polynomial vector fields of degree at most three; all other systems are treated as OOD for `FIM-ODE`. The OOD split refers to the pretraining distribution of `FIM-ODE` and does not imply that these systems are OOD for `ODEFormer`. Bold values indicate strict improvements over the other model for the same task, split, and corruption setting.

| Task | Split | Model | $(0, 0)$ | $(0, 0.03)$ | $(0, 0.05)$ | $(0.5, 0)$ | $(0.5, 0.03)$ | $(0.5, 0.05)$ |
|---|---|---|---|---|---|---|---|---|
| Reconstruction | 1D ID (11 ODEs) | `FIM-ODE` | **100.0** | **90.9** | 86.4 | **100.0** | 86.4 | **86.4** |
| | | `ODEFormer` | 86.4 | 86.4 | 86.4 | 86.4 | **90.9** | 77.3 |
| | 1D OOD (12 ODEs) | `FIM-ODE` | **100.0** | **100.0** | **100.0** | **100.0** | **100.0** | **95.8** |
| | | `ODEFormer` | 91.7 | 91.7 | 91.7 | 95.8 | 95.8 | 87.5 |
| | 2D ID (15 ODEs) | `FIM-ODE` | **93.3** | **93.3** | **86.7** | **90.0** | **83.3** | **76.7** |
| | | `ODEFormer` | 50.0 | 46.7 | 50.0 | 53.3 | 63.3 | 46.7 |
| | 2D OOD (13 ODEs) | `FIM-ODE` | **92.3** | **92.3** | **76.9** | **88.5** | **80.8** | **76.9** |
| | | `ODEFormer` | 69.2 | 65.4 | 61.5 | 61.5 | 53.8 | 73.1 |
| | 3D ID (9 ODEs) | `FIM-ODE` | **16.7** | 0.0 | 5.6 | **16.7** | 0.0 | 5.6 |
| | | `ODEFormer` | 11.1 | **11.1** | 5.6 | 11.1 | **16.7** | **11.1** |
| | 3D OOD (1 ODE) | `FIM-ODE` | 100.0 | 100.0 | 100.0 | 100.0 | 100.0 | 100.0 |
| | | `ODEFormer` | 100.0 | 100.0 | 100.0 | 100.0 | 100.0 | 100.0 |
| Generalization | 1D ID (11 ODEs) | `FIM-ODE` | 54.5 | **59.1** | 50.0 | 50.0 | 54.5 | 54.5 |
| | | `ODEFormer` | **59.1** | 54.5 | **54.5** | **63.6** | **63.6** | 54.5 |
| | 1D OOD (12 ODEs) | `FIM-ODE` | **50.0** | **50.0** | **50.0** | **45.8** | 37.5 | **54.2** |
| | | `ODEFormer` | 37.5 | 37.5 | 29.2 | 33.3 | **41.7** | 41.7 |
| | 2D ID (15 ODEs) | `FIM-ODE` | **26.7** | **36.7** | **30.0** | 20.0 | 23.3 | **23.3** |
| | | `ODEFormer` | 20.0 | 26.7 | 20.0 | **30.0** | **33.3** | 20.0 |
| | 2D OOD (13 ODEs) | `FIM-ODE` | 15.4 | 15.4 | 11.5 | 19.2 | 19.2 | 11.5 |
| | | `ODEFormer` | **19.2** | 15.4 | **19.2** | 19.2 | 19.2 | **15.4** |
| | 3D ID (9 ODEs) | `FIM-ODE` | 0.0 | 0.0 | 5.6 | 0.0 | 0.0 | 0.0 |
| | | `ODEFormer` | **5.6** | **5.6** | 5.6 | **11.1** | **5.6** | **5.6** |
| | 3D OOD (1 ODE) | `FIM-ODE` | 0.0 | 0.0 | 0.0 | 0.0 | 0.0 | 0.0 |
| | | `ODEFormer` | 0.0 | 0.0 | 0.0 | 0.0 | 0.0 | 0.0 |

*Table 11.* ODEBench results using MSE instead of the thresholded $R^2$ metric. We report the 0.05-quantile, median, and 0.95-quantile of the MSE for reconstruction and generalization. Results are reported separately by dimension to account for differences in scale across systems. Bold values indicate strict improvements over the other model for the same task, dimension, and metric.

| Task | Dimension | Model | 0.05-quantile MSE | Median MSE | 0.95-quantile MSE |
|------|-----------|-------|-------------------|------------|-------------------|
| Reconstruction | 1D | FIM-ODE | 1.36e-05 | **6.63e-03** | **3.29** |
| | | ODEFormer | **9.80e-06** | 1.03e-02 | 8.36 |
| | 2D | FIM-ODE | 7.16e-05 | **2.14e-02** | **2.22** |
| | | ODEFormer | **3.81e-05** | 8.55e-02 | 24.68 |
| | 3D | FIM-ODE | **5.64e-04** | **8.33** | 2492.35 |
| | | ODEFormer | 3.94e-03 | 13.79 | **1749.21** |
| Generalization | 1D | FIM-ODE | 6.31e-05 | 2.89e-01 | 859.37 |
| | | ODEFormer | **2.61e-05** | **2.16e-01** | **445.71** |
| | 2D | FIM-ODE | 1.11e-03 | **5.51e-01** | **40.88** |
| | | ODEFormer | **3.43e-04** | 8.36e-01 | 566.72 |
| | 3D | FIM-ODE | 1.23e-01 | **9.57** | 2575.50 |
| | | ODEFormer | **3.78e-02** | 15.18 | **2496.89** |

*Table 12.* Trajectory Generalization Task on ODEBench: Case $\{R^2 > 0.8\}$

| Method | $\rho = 0.0$ | | | $\rho = 0.5$ | | |
|--------|--------------|--------------|--------------|--------------|--------------|--------------|
| | $\sigma = 0.0$ | $\sigma = 0.03$ | $\sigma = 0.05$ | $\sigma = 0.0$ | $\sigma = 0.03$ | $\sigma = 0.05$ |
| ODEFormer | 35.2% | 32.8% | 34.4% | 35.2% | 41.0% | 33.6% |
| FIM-ODE | 41.8% | 41.8% | 38.5% | 36.1% | 40.2% | 39.3% |

### E.4. Fixed-point diagnostics.

Table 13 reports a local stability analysis of the vector fields inferred by FIM-ODE and ODEFormer. For learned models, candidate equilibria are obtained by minimizing $\|\hat{f}(x)\|$, and stability is classified from the eigenvalues of $\nabla_x \hat{f}(x^*)$. We report the residual $\|\hat{f}(x^*)\|$ because our neural vector field estimates need not attain exact zeros.

The diagnostics reveal information that is not visible from trajectory error alone. For the frictionless pendulum, the ground truth has a marginal centre at the origin and a saddle at the upright equilibrium. ODEFormer preserves this structure, whereas FIM-ODE turns the centre into weak unstable spirals, reflecting the difficulty of preserving conservative geometry with an unconstrained neural vector field. For CDIMA, FIM-ODE finds a nearby candidate equilibrium with the correct unstable-spiral type, although with a non-negligible residual and an overestimated growth rate. ODEFormer, in contrast, finds a near-exact symbolic equilibrium but reverses the local stability, predicting a stable spiral. For the Lotka–Volterra competition model, FIM-ODE partially recovers the richer topology by identifying both stable-node and saddle candidates, but the equilibrium locations and multiplicities are distorted. Neither method recovers the extinction equilibrium near $(0, 0)$, consistent with the fact that the context trajectories do not constrain this region of phase space.

These results support the main interpretability claim: although FIM-ODE does not output a closed-form symbolic expression, it provides a queryable vector-field representation that can be analysed with the same local tools used for symbolic ODEs. The analysis also exposes failure modes that trajectory reconstruction alone would hide.

*Table 13.* Full fixed-point diagnostics on selected ODEBench systems. For the ground-truth systems, $f(x^*) = 0$ exactly. For learned models, $x^*$ denotes a candidate equilibrium found by minimizing $\|\hat{f}(x)\|$, and the final column reports the corresponding residual. The Jacobian spectrum is summarized by $\max \Re(\lambda)$.

| System | Model | Candidate equilibrium $x^*$ | Stability | $\max \Re(\lambda)$ | $\|\hat{f}(x^*)\|$ |
|---|---|---|---|---|---|
| Pendulum 28 | GT | $(0.000, 0.000)$ | marginal centre | $+0.0000$ | $0$ |
| | | $(3.142, 0.000)$ | saddle | $+0.9487$ | $0$ |
| | FIM-ODE | $(-0.069, 0.047)$ | unstable spiral | $+0.0813$ | $2.59{\times}10^{-2}$ |
| | | $(0.154, -0.096)$ | unstable spiral | $+0.0956$ | $4.42{\times}10^{-2}$ |
| | | $(0.154, 0.154)$ | unstable spiral | $+0.1195$ | $1.08{\times}10^{-1}$ |
| | | $(-0.154, 0.154)$ | unstable spiral | $+0.1781$ | $1.27{\times}10^{-1}$ |
| | ODEFormer | $(-0.000, -0.000)$ | marginal centre | $+0.0000$ | $5.42{\times}10^{-9}$ |
| | | $(-3.110, -0.000)$ | saddle | $+1.3868$ | $1.11{\times}10^{-8}$ |
| | | $(4.919, -0.000)$ | saddle | $+1.9562$ | $2.04{\times}10^{-8}$ |
| CDIMA 42 | GT | $(1.780, 4.168)$ | unstable spiral | $+0.2415$ | $0$ |
| | FIM-ODE | $(1.692, 4.154)$ | unstable spiral | $+0.5879$ | $3.70{\times}10^{-1}$ |
| | | $(2.000, 4.462)$ | unstable spiral | $+0.5856$ | $5.25{\times}10^{-1}$ |
| | | $(1.385, 3.885)$ | stable spiral | $-0.5201$ | $6.42{\times}10^{-1}$ |
| | | $(2.000, 4.154)$ | unstable spiral | $+0.4724$ | $6.76{\times}10^{-1}$ |
| | ODEFormer | $(1.693, 4.640)$ | stable spiral | $-0.2698$ | $1.38{\times}10^{-8}$ |
| LV competition 26 | GT | $(0.000, 0.000)$ | unstable node | $+3.0000$ | $0$ |
| | | $(0.000, 2.000)$ | stable node | $-1.0000$ | $0$ |
| | | $(1.000, 1.000)$ | saddle | $+0.4142$ | $0$ |
| | | $(3.000, 0.000)$ | stable node | $-1.0000$ | $0$ |
| | FIM-ODE | $(0.769, 1.077)$ | saddle | $+0.0370$ | $2.15{\times}10^{-1}$ |
| | | $(0.462, 1.385)$ | saddle | $+0.1074$ | $2.15{\times}10^{-1}$ |
| | | $(-0.154, 2.000)$ | stable node | $-1.4701$ | $2.18{\times}10^{-1}$ |
| | | $(1.077, 0.462)$ | stable node | $-0.0623$ | $2.39{\times}10^{-1}$ |
| | ODEFormer | $(-0.000, 1.156)$ | saddle | $+0.2015$ | $1.47{\times}10^{-9}$ |
| | | $(-0.000, 0.982)$ | stable node | $-0.1561$ | $1.72{\times}10^{-9}$ |
| | | $(2.833, 0.010)$ | stable node | $-0.0001$ | $4.20{\times}10^{-7}$ |

# F. Additional Experiments and Results: VDP and FHN

### F.1. Effect of context sizes

In Section 5.2, we reported the largest-context result for `FIM-ODE` in the low-data VDP forecasting and FHN imputation experiments. Here, we provide the full context-size ablation. Starting from the fixed dataset of Hegde et al. (2022), we progressively increase the amount of context by adding noisy trajectories. We report results for $k \in \{3, 9, 50\}$ total context trajectories, corresponding to the original trajectory plus 2, 8, or 49 additional trajectories. For each value of $k$, we repeat the experiment over 100 independent noise realisations.

We consider two ways of sampling the initial conditions of the additional context trajectories. In the *perturbed* setting, which is the setting used in the main text, initial conditions are sampled from a normal distribution centred at the initial condition of the target trajectory, with variance $0.1$. In the *random* setting, initial conditions are sampled uniformly from a wider domain: $[-3, 3]^2$ for VDP and $[-2, 2]^2$ for FHN. All additional trajectories use the same observation time grid as the corresponding context dataset in the VDP case, whereas the trajectories of FHN have every observation that falls in the quadrant $x_1 > 0, x_2 < 0$ removed.

Table 14 reports the mean, standard deviation, median, and 5th/95th percentiles of the MSE.

The results show that increasing the number of context trajectories generally reduces the variability of zero-shot inference. This effect is clearest in the perturbed setting, which matches the main-text experiment: for VDP (T1), the standard deviation decreases from $0.3851$ at $k = 3$ to $0.0839$ at $k = 50$, while for FHN it decreases from $0.1772$ to $0.0664$. The random setting can further improve performance when the additional trajectories provide broader coverage of the relevant state space, as

*Table 14.* Effect of context size on zero-shot `FIM-ODE` performance in the low-data VDP forecasting and FHN imputation experiments. Lower is better. VDP (T1) denotes the uniformly spaced forecasting task, and VDP (T2) the irregularly sampled forecasting task. The column $k$ denotes the total number of context trajectories, including the original trajectory from Hegde et al. (2022); thus $k = 3, 9, 50$ correspond to adding 2, 8, and 49 additional trajectories. In the *perturbed* setting, additional initial conditions are sampled from a normal distribution centred at the target initial condition, with variance $0.1$. In the *random* setting, they are sampled uniformly from $[-3, 3]^2$ for VDP and $[-2, 2]^2$ for FHN. Results are computed over 100 independent noise realisations.

| Task | IC mode | $k$ | Mean | Std. | Median | Q05 | Q95 |
|------|---------|-----|------|------|--------|-----|-----|
| VDP (T1) | perturbed | 3 | 0.4730 | 0.3851 | 0.3821 | 0.0586 | 1.3269 |
| VDP (T1) | perturbed | 9 | 0.4474 | 0.2730 | 0.3958 | 0.1551 | 0.8743 |
| VDP (T1) | perturbed | 50 | 0.3581 | 0.0839 | 0.3514 | 0.2242 | 0.5041 |
| VDP (T1) | random | 3 | 0.2323 | 0.2843 | 0.1313 | 0.0304 | 0.6423 |
| VDP (T1) | random | 9 | 0.1673 | 0.1832 | 0.1050 | 0.0268 | 0.6674 |
| VDP (T1) | random | 50 | 0.0611 | 0.0299 | 0.0523 | 0.0329 | 0.1316 |
| VDP (T2) | perturbed | 3 | 1.3922 | 0.9678 | 1.1894 | 0.2011 | 3.2008 |
| VDP (T2) | perturbed | 9 | 0.9251 | 0.4891 | 0.8647 | 0.1702 | 1.6795 |
| VDP (T2) | perturbed | 50 | 0.8874 | 0.2422 | 0.8762 | 0.4961 | 1.3293 |
| VDP (T2) | random | 3 | 2.2462 | 0.8580 | 2.1763 | 0.7869 | 3.8837 |
| VDP (T2) | random | 9 | 2.3200 | 0.9387 | 2.1602 | 1.0047 | 4.1252 |
| VDP (T2) | random | 50 | 2.1298 | 0.3693 | 2.1203 | 1.5802 | 2.7659 |
| FHN | perturbed | 3 | 0.5110 | 0.1772 | 0.4832 | 0.3083 | 0.8726 |
| FHN | perturbed | 9 | 0.4196 | 0.1262 | 0.4092 | 0.2471 | 0.6510 |
| FHN | perturbed | 50 | 0.4358 | 0.0664 | 0.4229 | 0.3512 | 0.5449 |
| FHN | random | 3 | 0.4491 | 0.1803 | 0.4198 | 0.2162 | 0.7853 |
| FHN | random | 9 | 0.3511 | 0.1076 | 0.3359 | 0.1935 | 0.5310 |
| FHN | random | 50 | 0.2640 | 0.0491 | 0.2598 | 0.1717 | 0.3511 |

seen for VDP (T1) and FHN. However, this is not uniform across tasks: for VDP (T2), randomly sampled initial conditions lead to higher errors than nearby perturbations, suggesting that broader context is not necessarily useful when it is less aligned with the target forecasting trajectory. Overall, these results support the conclusion that `FIM-ODE` benefits from richer context, but that the usefulness of additional trajectories depends on how well they constrain the dynamics relevant to the target prediction problem.

## G. Synthetic Polynomial ODEs & Ablations: Comparison with `ODEFormer`

In this section, we provide detailed results comparing `FIM-ODE` against `ODEFormer` (d'Ascoli et al., 2024) on synthetic polynomial ODE systems, evaluating both in-distribution performance (degree-3 polynomials matching the pretraining distribution) and out-of-distribution generalization (degree-6 polynomials).

**In-Distribution Synthetic Polynomials.** We first evaluate on 4,000 synthetic polynomial ODE systems with maximum degree 3, matching the pretraining distribution. Figure 13 shows that `FIM-ODE` substantially outperforms `ODEFormer` across all dimensionalities in both reconstruction and generalization tasks. For reconstruction, `FIM-ODE` achieves 90% success rate (trajectories with $R^2 > 0.9$) compared to `ODEFormer`'s 65%. The generalization gap is even more pronounced: `FIM-ODE` reaches 26% while `ODEFormer` achieves only 18.5%. This advantage stems from `FIM-ODE`'s continuous neural operator representation, which naturally interpolates between observed states, whereas `ODEFormer`'s symbolic expressions may struggle with discrete token limitations.

**Out-of-Distribution Generalization.** To assess robustness beyond the training distribution, we evaluate both models on degree-6 polynomials which are well outside the degree-3 pretraining regime. Table 15 shows that while both models experience performance degradation, `FIM-ODE` maintains its advantage. Remarkably, in the single-trajectory reconstruction setting, `FIM-ODE`'s performance drops by only 2.9 percentage points ($90.2\% \rightarrow 87.3\%$), demonstrating surprising resilience to higher-order polynomial dynamics. `ODEFormer` shows a larger degradation of 5.5 points. For generalization tasks, both models experience larger drops, consistent with the inherently harder task of extrapolating to unseen initial conditions with more complex dynamics.

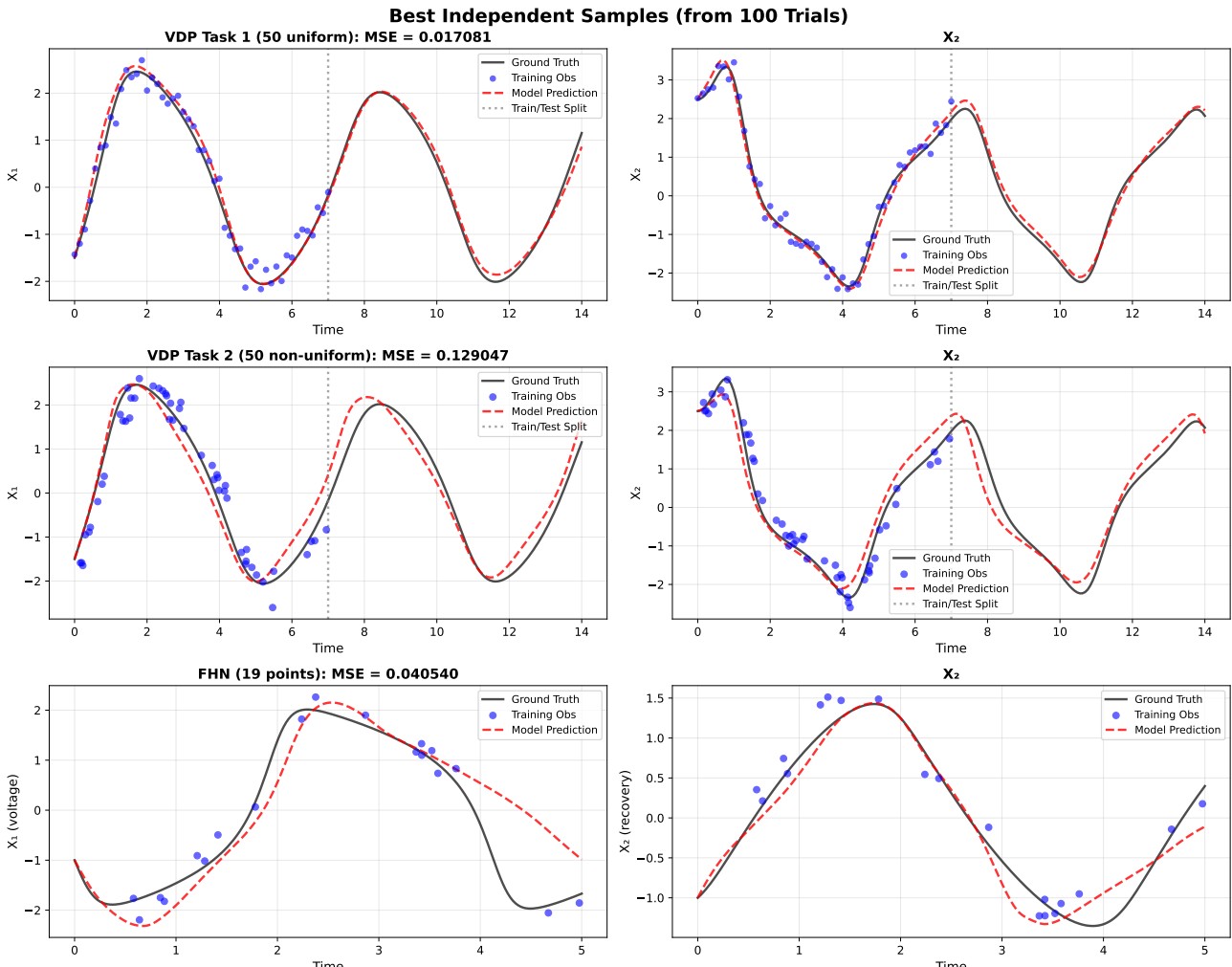

*Figure 12.* Best-performing random samples from 100 independent noise/sampling trials for each benchmark. Blue dots indicate training observations, black lines show ground truth, and red dashed lines show model predictions.

## G.1. Discretization Ablations

We evaluate `FIM-ODE`'s ability to handle different trajectory discretizations and varying numbers of input trajectories, testing its neural operator capabilities on in-distribution polynomial ODEs.

**Experimental Setup.** We test the model on 4,000 newly generated polynomial ODE systems (maximum degree 3) with varying numbers of trajectory points $n_{points} \in \{50, 100, 200, 250\}$ and varying numbers of input trajectories $K \in \{1, 9, 12\}$. The choice of $n_{points}$ spans the range seen during training (50–200 points) and extends beyond it (250 points) to test extrapolation. Similarly, $K$ ranges from the minimum (1) to maximum (9) seen during training, plus a test case beyond the training distribution (12 trajectories).

**Results: Discretization Effects.** Figure 14 shows reconstruction performance (percentage of trajectories with $R^2 > 0.9$) across all combinations of discretization levels and trajectory counts. For single-trajectory reconstruction, finer discretization provides marginal benefits. However, for 9 and 12 trajectories, performance slightly decreases as the number of points increases. This counter-intuitive result suggests that with more context trajectories available, the model can effectively interpolate the flow field even from coarser discretizations, and overly fine discretizations may introduce unnecessary complexity.

**Dimension-Specific Performance.** Figure 15 presents dimension-specific reconstruction results across all tested configurations. Each cell shows the percentage of $d$-dimensional trajectories achieving $R^2 > threshold$ for varying thresholds. A

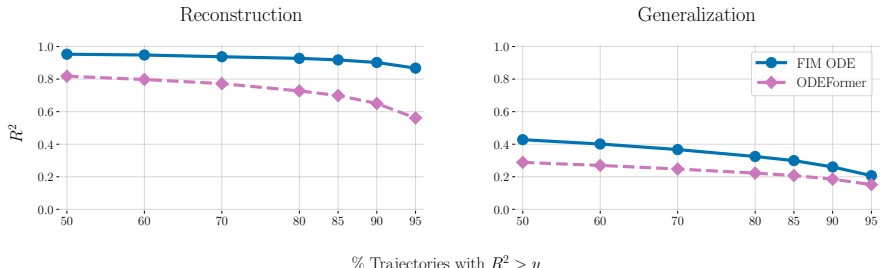

*Figure 13.* Comparison of `FIM-ODE` and `ODEFormer` on polynomial ODEs (degree $\leq 3$). Left: reconstruction performance. Right: generalization to new initial conditions. Both evaluated across dimensions $d \in \{1, 2, 3\}$.

*Table 15.* Out-of-distribution performance on degree-6 polynomials. Values show % of trajectories with $R^2 > 0.9$. Models were trained only on degree-3 polynomials.

| Method | Degree 3 | Degree 6 | $\Delta$ |
|---|---|---|---|
| *Reconstruction (1 trajectory)* | | | |
| `FIM-ODE` | 90.2 | 87.3 | $-2.9$ |
| `ODEFormer` | 65.0 | 59.5 | $-5.5$ |
| *Generalization (1 trajectory)* | | | |
| `FIM-ODE` | 26.0 | 15.0 | $-11.0$ |
| `ODEFormer` | 18.5 | 11.3 | $-7.2$ |

consistent pattern emerges: higher-dimensional trajectories are harder to reconstruct across all tested scenarios. This reflects the increased complexity of capturing coupled dynamics in higher dimensions and the curse of dimensionality in sampling the state space.

### G.2. Vector Field Quality Metrics

While trajectory-based evaluation is our primary metric, we also assess vector field prediction quality directly using RMSE and cosine similarity computed on the 10,000 vector field samples generated for each ODE system.

**Aggregated Results.** Figures 16 and 17 show vector field prediction quality across different discretizations and trajectory counts. Both metrics indicate that providing more input trajectories substantially improves vector field estimation. The number of trajectory points has a comparatively smaller impact than observed in trajectory reconstruction, likely because overall vector field error is dominated by regions with large magnitude, while fine details critical for accurate long-term trajectory integration have less weight in these aggregate metrics.

**Per-Dimension Analysis.** Figures 18 and 19 break down vector field quality by dimension for the standard 200-point discretization. An interesting observation is that while vector field predictions improve with more input trajectories, the benefit diminishes with increasing dimension. For 1D systems, the median RMSE decreases by a factor of approximately 4 when given 9 trajectories instead of 1. This improvement factor reduces to 1.8 in 2D and 1.3 in 3D. This suggests that higher-dimensional systems require substantially more trajectory coverage to fully constrain the flow field, a consideration for future work targeting higher dimensions.

### G.3. Impact of Dataset Size

We investigate how pretraining dataset size affects model performance by training four variants of `FIM-ODE` on datasets of 10,000, 50,000, 100,000, and 200,000 synthetic polynomial ODEs. To isolate the effect of dataset size, we use a smaller model architecture (approximately 1.3M parameters including uncertainty estimation) and train each variant for approximately 10 hours on a single NVIDIA A40 GPU.

**Polynomial ODE Performance.** Figure 20 shows reconstruction and generalization performance on in-distribution polynomial test data. Larger training datasets consistently produce better performing models, but improvements diminish beyond 100,000 training examples. This suggests diminishing returns for the tested model capacity, indicating that further

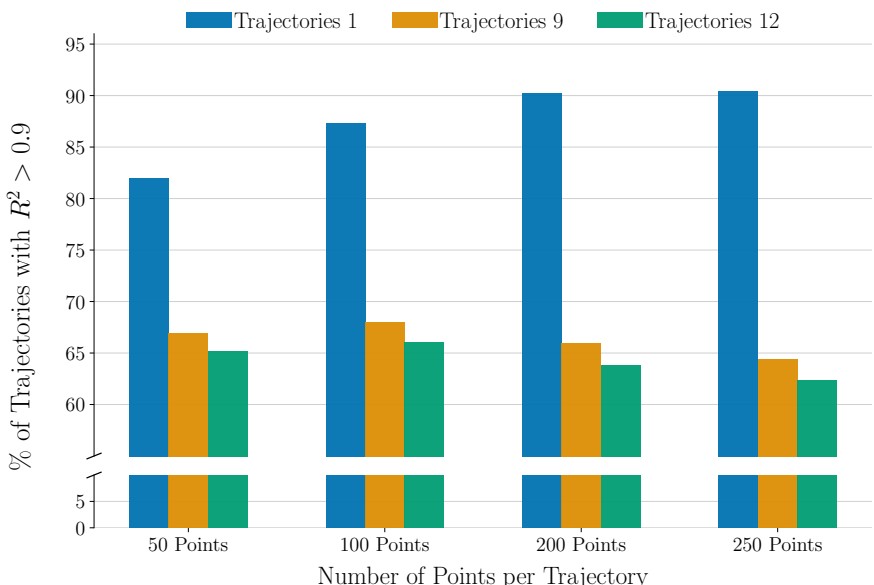

*Figure 14.* Reconstruction performance for different trajectory discretizations ($n_{points}$) and numbers of input trajectories ($K$). Higher is better.

scaling would require proportionally larger model architectures.

**ODEBench Performance.** Figures 21 and 22 show ODEBench performance across dataset sizes. For reconstruction, models trained on 200,000 systems perform comparably to those trained on 50,000 or 100,000 systems, with only minimal differences on uncorrupted inputs. This suggests that even moderate-scale pretraining (50,000–100,000 systems) provides sufficient coverage of polynomial ODE dynamics for zero-shot transfer to real-world systems. For generalization, larger datasets provide more consistent benefits, particularly under data corruption.

**Implications.** These results demonstrate that `FIM-ODE`'s foundation model approach benefits from scale, but also that moderate-scale pretraining (on the order of 100,000 diverse systems) is sufficient for strong zero-shot performance. The diminishing returns suggest that future improvements should focus on jointly scaling both data and model capacity, following established neural scaling laws.

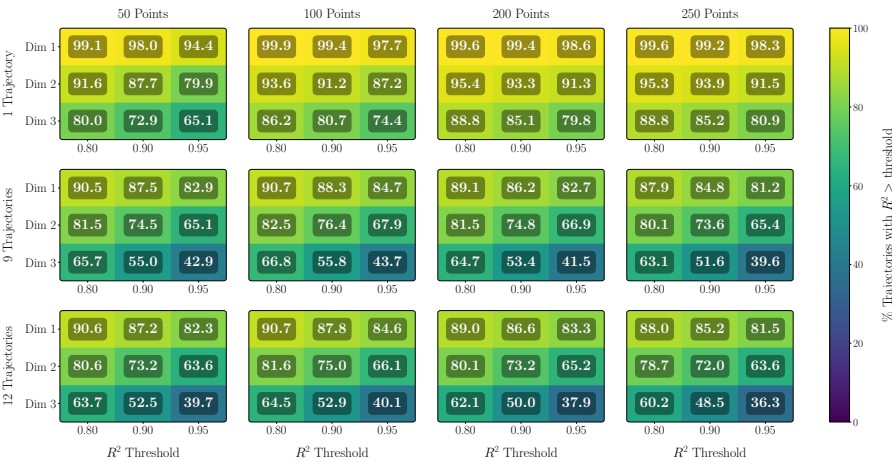

*Figure 15.* Reconstruction performance as percentage of trajectories with $R^2 > threshold$, shown separately for each dimension. Brighter colors indicate better performance. Results shown for all tested model inputs (combinations of $n_{points}$ and $K$).

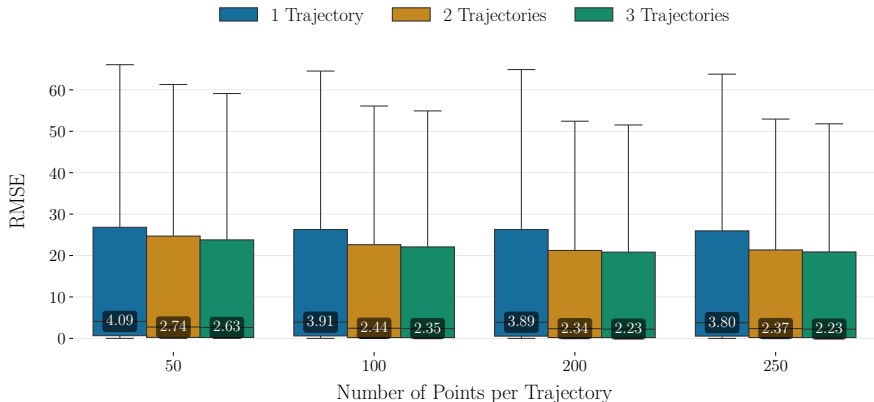

*Figure 16.* RMSE of vector field predictions for different trajectory discretizations and numbers of input trajectories. Lower is better.

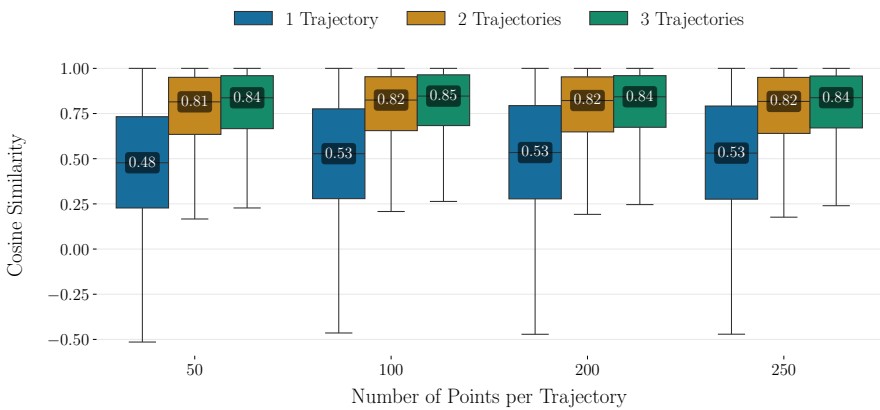

*Figure 17.* Cosine similarity between predicted and true vector fields for different trajectory discretizations and numbers of input trajectories. Higher is better.

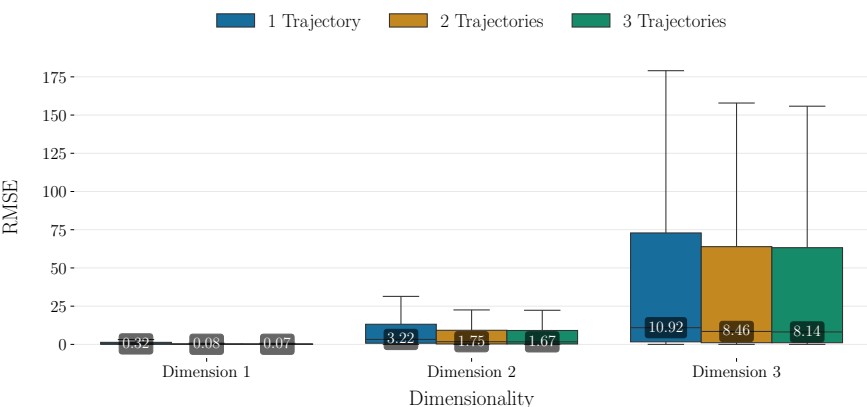

*Figure 18.* RMSE of vector field predictions per dimension for 200-point trajectory discretization. Lower is better.

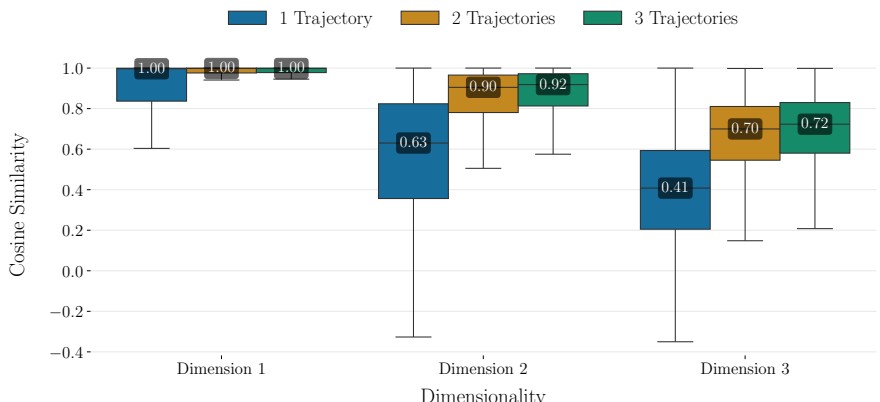

*Figure 19.* Cosine similarity between predicted and true vector fields per dimension for 200-point trajectory discretization. Higher is better.

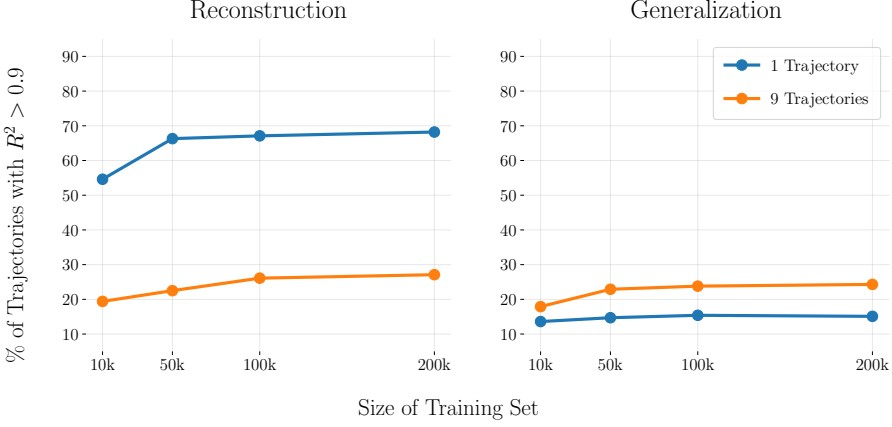

*Figure 20.* Reconstruction (left) and generalization (right) performance as a function of training dataset size. Performance measured as percentage of trajectories with $R^2 > 0.9$ on in-distribution polynomial ODEs.

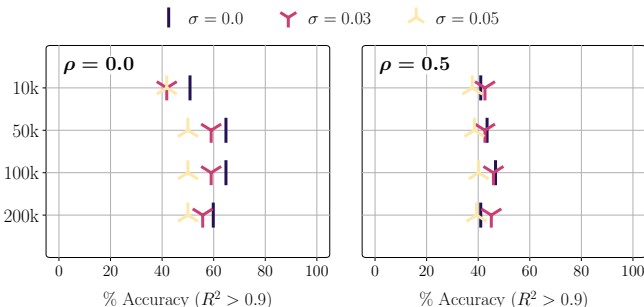

*Figure 21.* ODEBench reconstruction performance for models trained on 10k, 50k, 100k, and 200k polynomial ODEs. $\sigma$ denotes Gaussian noise and $\rho$ denotes the dropout ratio used for irregular sampling.

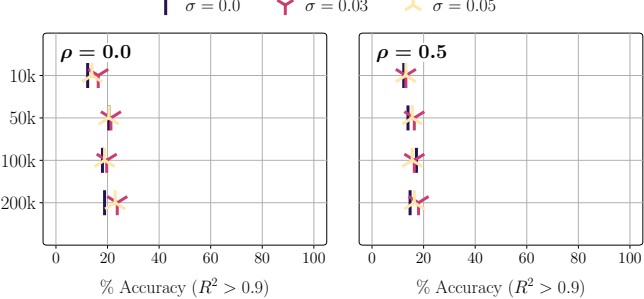

*Figure 22.* ODEBench generalization performance for models trained on 10k, 50k, 100k, and 200k polynomial ODEs. $\sigma$ denotes Gaussian noise and $\rho$ denotes the dropout ratio used for irregular sampling.

