# OpenReview forum: "Foundation Inference Models for Ordinary Differential Equations"
_ICML.cc/2026/Conference — ICML 2026 regular_

### Official Review · Reviewer_hxc2 · 2026-02-24

**Soundness:** 2
**Presentation:** 3
**Significance:** 2
**Originality:** 2
**Overall Recommendation:** 3
**Confidence:** 5

**Summary:**

The paper introduces FIM-ODE, a pretrained foundation model for amortised inference of ordinary differential equations (ODEs) from noisy, sparse trajectory data. The model is pretrained on a synthetic dataset of polynomial ODEs of degree at most three in up to three dimensions, using a transformer-based neural operator with an encoder-decoder cross-attention architecture.

**tl;dr** ODE manifolds were introduced a long time ago, it seems that authors accidentaly reinvent them

**Compliance With Llm Reviewing Policy:**

Affirmed.

**Final Justification:**

We put things simply. ODE inference is a long-standing research area and it should be surveyed as a whole to understand how and when foundational models add value. It is too bold to claim a foundation model for ODE inference (in other words, a foundation model for the initial value problem) based on the evidence presented.

The paper has a single major flaw. The manifold identification issue is not a matter of perspective -- it is a functional characterisation of what the method computes. A model trained exclusively on degree-3 polynomial ODEs develops an inductive bias toward the polynomial coefficient manifold regardless of whether the loss is written in function value space or coefficient space. These are two representations of the same fitting problem when the hypothesis class is fixed. The authors' own citation of Kawata et al. (2025) supports this reading: transformers select algorithms based on training data distribution, meaning the algorithm implicitly learned here is polynomial coefficient identification. Operator Inference solves the same problem via least squares. No evidence is provided that FIM-ODE generalises in ways a structured polynomial fit cannot, and the rebuttal does not engage with this operationally.

We find no merit in the foundational model framing in this form and believe the score accurately reflects the paper's contribution.

**Key Questions For Authors:**

1. How does FIM-ODE compare architecturally and empirically to Poseidon (Herde et al., 2024), which uses a near-identical cross-attention encoder-decoder design pretrained on diverse synthetic PDE data for zero-shot transfer? A direct comparison or a clear argument for why this comparison is not meaningful is necessary to assess the novelty claims. A positive response with experimental results would significantly change the evaluation.

2. On the VDP forecasting and FHN imputation tasks, how does FIM-ODE compare to modern pretrained zero-shot time series models such as Chronos or AutoGluon? Since all evaluation uses trajectory MSE, this comparison is necessary to establish whether vector field recovery provides any practical advantage over direct forecasting. If FIM-ODE is competitive, this would substantially strengthen the paper.

3. Can the authors provide a theoretical justification for why polynomial degree-3 priors generalise to out-of-distribution systems? Specifically, is the generalisation related to the compactness of the polynomial ODE coefficient manifold and the implicit local Taylor approximation of arbitrary smooth vector fields in the observed region? Making this argument explicit would substantially strengthen the theoretical contribution and could meaningfully change the evaluation.

4. Is there a concrete use case where recovering the vector field itself, as opposed to forecasting the trajectory, provides demonstrably added value? For example, downstream tasks such as control, parameter estimation, or model discovery would make a compelling case. Without such a demonstration, the motivation for the vector field recovery framing over direct forecasting is unclear.

5. Given that the architecture closely follows FIM-SDE (Seifner et al., 2025a), what are the specific novel architectural contributions of FIM-ODE beyond the adaptation to the deterministic ODE setting? How does this work distinguish itself from a straightforward application of the FIM framework to a new system class?

6. How does FIM-ODE compare to Operator Inference (OpInf, Peherstorfer and Willcox, 2016) and Koopman-based methods such as EDMD or deep Koopman variants (Lusch et al., 2018)? OpInf in particular shares the same structural assumption of low-degree polynomial vector fields and performs inference via least squares, making it a natural and computationally inexpensive baseline. If FIM-ODE does not outperform OpInf on low-dimensional polynomial systems, the justification for the deep learning approach requires a stronger argument. A response including this comparison would substantially affect the evaluation.

**Limitations:**

The authors have partially addressed limitations. They acknowledge the three-dimensional restriction, the non-stationary scaling of the polynomial prior, and the fixed-dimensionality architecture. However, the paper does not discuss the fundamental positioning issue between the symbolic regression and forecasting communities, nor does it acknowledge the absence of comparisons with neural operator baselines and pretrained time series models. A more complete limitations discussion would strengthen the paper.

**Strengths And Weaknesses:**

**General Strengths**

- The core empirical finding that a simple polynomial prior over low-degree ODEs is sufficient for competitive zero-shot inference is genuinely interesting and challenges the assumption that richer pretraining distributions are necessary for generalisation.
- The local neural operator framing provides a well-motivated alternative to global symbolic regression, and the comparison against ODEFormer is fair and clearly presented.
- The finetuning methodology is practically elegant: adapting the pretrained model as a neural ODE initialisation is lightweight, requires no ML expertise, and is shown to be effective even in very low-data regimes.
- Pretrained weights are released, lowering the barrier for scientists to apply the method without substantial ML infrastructure.
- The transition feature representation (state, displacement, squared displacement, inter-observation time) is a sensible and well-motivated ODE-specific inductive bias.

**General Weaknesses**
- The paper presents its encoder-decoder cross-attention architecture as a novel neural operator design, yet makes no reference to closely related work. Most notably, Poseidon (Herde et al., 2024) employs a near-identical architectural design: a perceiver-style encoder producing a permutation-invariant context representation followed by cross-attention from query locations, pretrained on diverse synthetic PDE data for zero-shot transfer. The absence of this comparison, both in related work and in experiments, is a significant omission that undermines the novelty claims. Physics-Informed Neural Operators (PINO) and FNO-based approaches are also not discussed despite addressing closely related function-to-function inference problems.
- The experimental evaluation on forecasting and imputation tasks uses only classical non-parametric baselines and ODEFormer. Modern pretrained zero-shot time series models such as Chronos, Moirai, TimesFM, or AutoGluon are not considered, despite being directly applicable and known to perform strongly in low-data regimes. Since all evaluation is trajectory-based MSE, these comparisons are necessary to establish whether vector field recovery provides any practical advantage over pure forecasting.
The paper falls between two communities without satisfying either. The symbolic regression and dynamical systems identification community would ask: where is interpretability? The paper recovers a neural vector field, not a symbolic equation, so no mechanistic insight into the underlying dynamics is provided. The time series forecasting community would ask: why not use a pretrained forecaster directly? The evaluation does not demonstrate any scenario where recovering the vector field itself provides added value over simply forecasting the trajectory. This fundamental positioning issue is not acknowledged by the authors.
- The most compelling explanation for why a degree-3 polynomial prior generalises to out-of-distribution systems is never articulated. Polynomial ODEs of degree at most three in at most three dimensions form a compact, finite-dimensional manifold in coefficient space. A model pretrained on sufficient coverage of this manifold can identify the locally best-fitting polynomial approximation to any smooth vector field in the region visited by the observed trajectory, essentially performing an implicit local Taylor approximation. This argument naturally explains the scaling results (diminishing returns beyond 100K systems once manifold coverage is sufficient), the OOD generalisation to non-polynomial systems, and the advantage of the local estimator. The paper gestures toward locality and the GP interpretation but never makes this argument explicit, which would significantly strengthen the theoretical contribution.
Architectural novelty is overstated. The architecture is closely derived from the prior work FIM-SDE (Seifner et al., 2025a), upon which this paper heavily builds. The adaptations for ODEs are reasonable engineering choices but do not constitute a novel architectural contribution. The relationship to FIM-SDE is underemphasised and the broader neural operator literature, particularly Poseidon, is not engaged with.
- The paper entirely omits a substantial and directly relevant family of ODE inference tools from the scientific computing community. Operator Inference (OpInf, Peherstorfer and Willcox, 2016) infers polynomial vector fields from trajectory data via least squares, using explicitly low-degree polynomial bases -- the exact same structural assumption underlying FIM-ODE's pretraining prior. This connection is close enough that the omission is surprising. Koopman operator methods, including DMD, EDMD, and deep Koopman variants (Lusch et al., 2018), similarly perform data-driven dynamical system identification from trajectory data, handle noisy and sparse observations, and provide principled theoretical frameworks for nonlinear dynamics. For low-dimensional polynomial systems, it is unclear whether FIM-ODE meaningfully outperforms a simple least-squares OpInf fit, which would call into question the justification for the deep learning machinery. These methods should be discussed in related work and ideally included as baselines.
- The restriction to at most three state dimensions is a fundamental limitation for real-world scientific modelling. The MoCap experiment, where five-dimensional data is projected to three dimensions with known information loss, illustrates rather than mitigates this problem.

## Soundness

**Strengths**

- The experimental results reported in the paper are internally consistent and the methodology is appropriate for the stated goals.
- The transition feature design, uncertainty weighting loss, and normalisation schemes are technically well-motivated and correctly implemented.
- The ablation studies on dataset size and discretisation are thorough and support the claims about scaling behaviour.

**Weaknesses**

- The experimental scope is too narrow to support the paper's broader claims about the benefits of local estimation and simple priors. Without comparisons to Poseidon, PINO, or pretrained forecasting models, the central empirical claims cannot be properly contextualised.
- The evaluation metrics are purely trajectory-based MSE. This does not test the quality of the recovered vector field in any scenario where the vector field itself matters, such as control, parameter estimation, or model discovery. The claimed contribution of vector field recovery is therefore not adequately supported by the evidence.
- No comparison is made with Operator Inference (OpInf) or Koopman-based baselines, which share the polynomial structural assumption and are computationally inexpensive. Without this, it is unclear whether the deep learning machinery provides any advantage over classical structured regression for the systems considered.
- The sensitivity analysis for the VDP and FHN experiments (Appendix E.2) reveals that the reported results depend strongly on the specific noise realisation. Reporting only the fixed benchmark seed in the main table without adequate discussion of this variability in the main text is misleading.



## Presentation


**Strengths**

- The paper is clearly written and the exposition of the method is easy to follow.
- The figures are informative and the appendices are thorough, providing sufficient detail for reproducibility.
- The distinction between trajectory reconstruction and trajectory generalisation tasks is well-drawn and the benchmark protocol is clearly described.

**Weaknesses**

- The related work section is notably narrow. It engages primarily with symbolic regression and classical non-parametric baselines while omitting the neural operator literature (Poseidon, PINO, FNO), pretrained time series models (Chronos, Moirai, TimesFM), and the operator inference and Koopman operator literature (OpInf, DMD, EDMD, deep Koopman) that are all directly relevant. This means the paper does not properly position itself relative to the field.
- The relationship to FIM-SDE (Seifner et al., 2025a), from which the architecture is closely derived, is underplayed. A reader unfamiliar with that work may overestimate the architectural novelty of the contribution.
- The paper does not acknowledge the fundamental tension in its positioning between the dynamical systems identification and time series forecasting communities. The introduction and conclusion would benefit from a clearer statement of the specific use case where the method is preferable to alternatives.


## Significance


**Strengths**

- The practical tool, a pretrained model for zero-shot ODE inference with released weights, has genuine utility for scientists working with low-dimensional dynamical systems who lack ML expertise.
- The finetuning methodology as a neural ODE initialisation strategy is practically appealing and could be adopted independently of the broader paper.

**Weaknesses**

- The three-dimensional limitation substantially restricts applicability to real scientific problems, most of which involve higher-dimensional state spaces.
- The evaluation does not demonstrate added value over existing pretrained forecasting tools. Without such a demonstration, it is unclear whether the vector field recovery framing provides any practical benefit over simpler approaches.
- The paper is unlikely to influence the broader machine learning community. Its scope is specialised, and the contribution does not provide the theoretical grounding or methodological advances that would make it a reference point for future work in scientific ML or neural operators.



## Originality

**Strengths**

- The observation that simple polynomial priors suffice for competitive amortised ODE inference is a genuinely interesting empirical finding that has not been clearly established before.
- The application of the FIM framework to ODEs, with ODE-specific transition features and uncertainty weighting, represents a coherent and well-executed instantiation of an existing framework.

**Weaknesses**

- The architectural design closely follows both FIM-SDE (Seifner et al., 2025a), a closely related prior work upon which this paper heavily builds, and Poseidon, an external concurrent work, without adequate acknowledgment or differentiation. The combination of cross-attention encoder-decoder with neural operator pretraining is not novel in this paper.
- The core finding about simple priors is presented as a surprising empirical result rather than a theoretically grounded insight. Without the polynomial manifold argument, the finding is an observation without an explanation, which limits its value as a scientific contribution.
- The paper does not introduce a new task, dataset, evaluation protocol, or theoretical framework. The contribution is an application of existing ideas to a specific problem class, which is a legitimate form of originality at ICML but requires stronger differentiation from related work than is currently provided.

---

> ### Author Rebuttal · Authors · 2026-03-31
>
> We thank the reviewer for the feedback. Several concerns revolve around fundamental distinctions central to our paper, namely **ODE inference vs. PDE operator learning**, **system identification vs. time-series forecasting**, **ODE vs. SDE inference**.
>
> To avoid repetition, we address these points collectively.
>
> **W1 / Q1.** We do **not** view Poseidon, PINO, or FNO as direct empirical baselines. These methods target PDE operator learning, whereas our task is *low-dimensional ODE system identification from noisy trajectories*. Our use of the term “neural operator” follows the FIM-SDE sense: FIM-ODE maps $K$ partially observed functions of time to another function, namely the target vector field. We agree that this distinction should be made clearer, and we will add an appendix discussion of neural-operator methods, explicitly including Poseidon.
>
> **W2.a / Q2.** We do **not** agree that pretrained forecasters such as Chronos or TimesFM are necessary baselines for our main claim. These models target forecasting, not system identification. This distinction matters: autonomous ODE models are constrained to explain the data through a time-homogeneous dynamical law. By contrast, direct forecasters may learn arbitrary, potentially inhomogeneous transition rules.
>
> Using MSE or $R^2$ does not erase this distinction. We adopt them solely for fair comparison with our ODE inference baselines.
>
> **W2.b / Q4 / W7 / W12.** System identification interpretability stems from the recovered model class (here, an ODE vector field), a mechanistic object forecasters lack. FIM-ODE yields inspectable vector fields (Fig. 2) that can be queried off-trajectory to e.g., recover fixed points and probe local stability via Jacobian eigenvalues. To make this explicit, we computed vector-field RMSE for FIM-ODE and ODEFormer on a grid covering the full VDP/FHN data (Hegde et al., 2022).
>
> ||VDP1|VDP2|FHN|
> |-|-|-|-|
> |FIM|2.73|2.81|3.27|
> |ODEF|4.01|4.58|4.13|
>
> These analyses directly show the value of vector-field recovery beyond forecasting.
>
> **W3.a / Q3.** Our motivation for combining a local representation with a polynomial prior is the intuition that low-dimensional polynomial systems provide useful **local** approximations to more general smooth vector fields in the region supported by the data. We appreciate the reviewer’s interpretation in terms of “coefficient manifolds”. We will add a discussion along these lines, while leaving a proper formalization to future work.
>
> **W3.b / Q5 / W11.** While we adapt the FIM-SDE architecture (line 230), **ODE inference is fundamentally different**. SDEs enjoy stronger identifiability guarantees because their stochastic component probes the entire state space (Bellot et al., 2021; Wang et al., 2023). Because ODEs lack this feature, prior design becomes significantly more delicate. Therefore, our work is not a straightforward transfer, but a nontrivial extension to a setting with different identifiability challenges.
>
> **References**
> - Bellot et al., 2021. Neural graphical modelling in continuous time: consistency guarantees and algorithms.
> - Wang et al., 2023. Neural structure learning with stochastic differential equations.
>
> **W4.a / Q6 / W8 / W10.** We do **not** view OpInf or Koopman methods as direct baselines. OpInf targets reduced-order modeling of *PDE-derived systems*, while Koopman methods learn lifted linear dynamics in an *observable space*. Though dynamically related, neither is a relevant baseline for low-dimensional ODE inference. We will explicitly discuss both in the appendix.
>
> **W4.b / W5 / W13 / W14–W18.** While our related work is already extensive **within the ODE inference community**, we will expand the appendix to discuss adjacent literatures for a broader ICML audience.
>
> We acknowledge the $d \leq 3$ limitation, as stated in our paper. However, low-dimensional ODEs are not merely niche: many important scientific systems admit low-dimensional descriptions, and our MoCap experiment demonstrates mapping high-dimensional observations to low-dimensional latent dynamics.
>
> Finally, *regarding VDP/FHN sensitivity*, this issue is not specific to FIM-ODE. ODEFormer exhibits similar sensitivity. We attribute this to the low-data regime, where neither model receives sufficient context to reliably *zero-shot estimate* the vector field away from the data. To support this, we repeated Experiment 2, adding trajectories with random initial conditions sampled from a normal distribution (variance 0.1) around the original ones.
>
> |#Traj.|VDP(T1)Med.|VDP(T1)0.95q|VDP(T2)Med.|VDP(T2)0.95q|FHNMed.|FHN0.95q|
> |---|---:|---:|---:|---:|---:|---:|
> |1|0.3792|1.9808|2.2443|4.9921|0.3263|1.1202|
> |2|0.3014|1.4856|1.4049|3.7391|0.2271|0.7691|
> |3|0.2617|1.4982|1.1946|3.3299|0.1931|0.5267|
> |9|0.3072|0.9138|1.0498|2.2369|0.1520|0.2417|
>
> The results demonstrate that increasing the amount of context substantially stabilizes inference, improving the Median MSE (Med) and reducing the 0.95-quantile of MSE.

---

> > ### Author Rebuttal · Reviewer_hxc2 · 2026-03-31
> >
> > ## Preliminary Remark: The Manifold Identification View
> >
> > Before addressing individual points, we articulate a unifying perspective that sharpens several concerns. The experimental setup can be described precisely as: given noisy sparse observations, identify the corresponding point in the finite-dimensional manifold of degree-3 polynomial ODE coefficients, then integrate the recovered global ODE for inference. This framing is more principled than the local estimator framing the authors adopt and has direct consequences for evaluation. Most importantly, it makes clear that Operator Inference (OpInf, Peherstorfer and Willcox, 2016), which identifies this same manifold point via least squares, is the most natural direct baseline. It also raises a fundamental question the paper does not address: given that manifold point identification is well-posed and efficiently solvable by classical structured regression, what does the deep learning amortisation actually buy? The rebuttal does not engage with this framing, and several responses are harder to defend in light of it.
> >
> >
> > ## Response to Individual Rebuttal Points
> >
> > **W1 / Q1 (Poseidon, PINO, FNO).** The authors' response deflects the concern rather than addressing it. We did not claim that Poseidon is a baseline for ODE system identification. We claimed that the cross-attention encoder-decoder pretraining paradigm is architecturally near-identical to Poseidon regardless of whether the domain is ODE or PDE. The authors promise an appendix discussion but no experimental comparison. This does not resolve the originality concern. We maintain this weakness.
> >
> > **W2.a / Q2 (Forecasting baselines).** The authors' conceptual argument that autonomous ODE models are constrained to time-homogeneous dynamical laws while forecasters are not is a legitimate distinction. We accept this as a partial response. However, the practical consequence of this constraint for the benchmark tasks (VDP forecasting, FHN imputation) is never quantified. Knowing that the ODE constraint helps or hurts relative to unconstrained forecasters would be informative regardless of the conceptual distinction.
> >
> > **W2.b / Q4 (Vector field interpretability).** The new VDP/FHN vector field RMSE table is a useful addition and the Jacobian eigenvalue and fixed point analysis is a legitimate demonstration of added value beyond forecasting. We accept this as a genuine improvement, though the comparison remains limited to ODEFormer rather than classical baselines.
> >
> > **W3.b / Q5 (ODE vs SDE distinction).** The identifiability argument is technically sound: ODEs are harder to identify than SDEs because the stochastic component provides richer state space coverage. We accept this as a legitimate distinction we did not credit sufficiently.
> >
> > **W3.a / Q3 (Manifold and polynomial prior justification).** The authors acknowledge our coefficient manifold interpretation and agree to add a discussion. However, they explicitly defer formalisation to future work. Given that this argument is the strongest available justification for the paper's central empirical finding, leaving it informal weakens the theoretical contribution. We encourage the authors to go further than a discussion paragraph.
> >
> > **W4.a / Q6 (OpInf and Koopman).** This is the weakest point in the rebuttal. The authors state that OpInf targets "reduced-order modelling of PDE-derived systems." This characterisation is inaccurate. OpInf (Peherstorfer and Willcox, 2016) was explicitly designed to learn polynomial ODE vector fields from trajectory data, which is precisely the setting of this paper. Under the manifold identification view articulated above, OpInf is the most direct classical baseline possible: it finds the polynomial coefficient manifold point via least squares. The dismissal of OpInf without engaging with its actual formulation is a significant weakness in the rebuttal and raises a concern about the authors' familiarity with the relevant literature. We maintain this as an unresolved weakness.
> >
> > **VDP/FHN sensitivity.** The multi-trajectory stabilisation table is informative. However, the main text still reports single-seed results without adequate discussion of variability. The rebuttal confirms rather than resolves the concern.
> >
> >
> > ## Updated Assessment
> >
> > The rebuttal partially improves the paper in two respects: the vector field RMSE evidence (W2.b) and the ODE/SDE identifiability distinction (W3.b) are genuine contributions. However, the two most fundamental concerns remain unaddressed. The Poseidon architectural similarity is deflected without experimental engagement, and the OpInf dismissal is factually inaccurate in light of the manifold identification view. We maintain the Reject recommendation. The score could move to 3 (weak reject) in recognition of the partial improvements, but the paper requires substantial revision before it can be meaningfully built upon by others.

---

> > > ### Author Response · Authors · 2026-04-06
> > >
> > > **@Preliminary remark / W3.A / Q3**: As we previously stated, we appreciate the reviewer’s interpretation in terms of a “coefficient manifold.” However, this is *not* the perspective we adopt, nor do we believe it is the most appropriate way to understand our proposal. The reason is simple:
> > >
> > > FIM-ODE is not trained to identify a point on the manifold of degree-3 polynomial coefficients. Rather, as is clear from our loss function, the model is trained explicitly to *match the values of the vector field in the region of state space covered by the context data*. These estimates are, of course, independent of any particular polynomial representation, and in our view this is precisely what enables OOD generalization beyond polynomial data. What matters for generalization is the *variety of local patterns* in the space of function values seen during training, rather than the polynomial representation itself (see e.g. Kawata et al. 2025).
> > >
> > > That said, we are happy to acknowledge the reviewer’s proposed “coefficient manifold” interpretation in the appendix as an alternative perspective on OOD generalization.
> > >
> > > **@W1/Q1**: We honestly do not understand how POSEIDON could be considered *architecturally near-identical* to FIM-ODE. POSEIDON is described by its own authors as a multiscale vision transformer built from SwinV2 blocks with shifted-window attention, patch merging and expansion, and lead-time-conditioned normalization. Its inputs are spatial fields treated as images, which are first partitioned into patches and then processed through a hierarchical U-Net-like encoder-decoder.
> > >
> > > By contrast, FIM-ODE’s encoder processes trajectory-derived input features
> > >
> > > $\{\mathbf{y}_i, \Delta \mathbf{y}_i, \Delta \mathbf{y}_i^2 \Delta \tau\}$,
> > >
> > > (which are extracted from the observed trajectories $\mathbf{y}_1, \dots, \mathbf{y}_L$), using linear self-attention, followed by a "cross-attention" decoder that uses the evaluation location of the target function $\mathbf{x}$ as queries.
> > >
> > > The reviewer also states that they do not view POSEIDON as a baseline for ODE system identification. This makes it unclear to us what form of *experimental comparison* is being requested, given that the two models are designed for substantially different input modalities and learning objectives.
> > >
> > > **@W4.a / Q6**: The reviewer’s emphasis on this comparison appears to follow directly from their proposed “coefficient manifold” interpretation. As stated above, this is not the perspective under which we formulated or trained FIM-ODE. Nevertheless, we would be happy to adapt OpInf to our setup and test its performance against our method.
> > >
> > > We also hope the reviewer does not lose sight of the main contribution of our paper: *the amortization of dynamical system inference*.
> > >
> > > **@VDP/FHN sensitivity**: We cannot modify the main text during the rebuttal phase. In the revision, we will update the main text accordingly to reflect the new results.
> > >
> > > **References**
> > >
> > > - Kawata et al. (2025): From Shortcut to Induction Head: How Data Diversity Shapes Algorithm Selection in Transformers

---

### Official Review · Reviewer_7N6t · 2026-03-07

**Soundness:** 3
**Presentation:** 3
**Significance:** 2
**Originality:** 2
**Overall Recommendation:** 3
**Confidence:** 3

**Summary:**

The paper introduces FIM-ODE, a very simple pretrained foundation inference framework for estimating the vector fields of ODEs from noisy, irregularly-sampled trajectory data. The model predicts a continuous local vector field in a single forward pass instead of solving a per-dataset optimization problem. The authors pretrain the model using a synthetic prior distribution of low-degree multivariate polynomial ODEs spanning 1-3 dimensions. Using a transformer-based neural operator architecture, the model learns to map trajectory contexts into a continuous, local functional representation of the vector field. Through empirical evaluations on synthetic benchmarks and real-world datasets, the authors demonstrate that FIM-ODE achieves strong zero-shot inference, achieving comparable or superior performance over global symbolic regression baselines like ODEFormer. The paper shows that the pretrained weights could serve as a highly effective initialization for rapid finetuning on complex, out-of-distribution dynamics.

**Compliance With Llm Reviewing Policy:**

Affirmed.

**Key Questions For Authors:**

1. While Appendix E.5 provides aggregate RMSE and Cosine Similarity for the vector fields, do you have spatial breakdowns of these errors? Understanding where the vector field estimation is most accurate would strengthen the claim that FIM-ODE is performing robust inference rather than just matching trajectories.
2. How sensitive are ODEBench and finetuning results to the choice of numerical solver and step size during evaluation? Would the relative ranking change under higher-order solvers or tighter tolerances?
3. Can you provide ablations that isolate the contribution of transition features, uncertainty weighting, and multi-trajectory conditioning to the large reconstruction gains? More explicitly, what drives the ODEBench reconstruction gap?
4. Beyond the proposed future axial attention direction, have you tested intermediate approaches like block-wise factorized encoders or latent embeddings that could extend beyond 3D without changing the entire data-generation pipeline?
5. Could you provide the raw, continuous error metrics (e.g., median MSE across all benchmark systems) rather than heavily relying on the arbitrary R^2 > 0.9 threshold?

**Limitations:**

A revelant work should be compared

[1] Jeffrey B. Lai, Anthony Bao, William Gilpin. Panda: A pretrained forecast model for universal representation of chaotic dynamics. ICLR, 2026.

**Strengths And Weaknesses:**

**Strengths:**
1. The method design is technically coherent, representing inference as a context-conditioned operator and using transition features (state, displacement, squared displacement, and time gaps), which are well-motivated for irregularly sampled trajectories and local flow estimation.
2. The training objective addresses heteroscedasticity across state space via a learned uncertainty head and a Laplace-style weighted loss, which is reasonable for preventing large-magnitude regions from dominating training. Also, the paper explicitly normalizes state dimensions and time gaps, then maps predictions back via a chain-rule transformation, which is thoughtful because ODE magnitudes vary widely across systems.
3. The idea of amortized inference for ODE identification is important. Reducing per-instance optimization could meaningfully broaden usage in engineering if robust. Also, the result that a much simpler pretraining prior can still transfer to OOD vector fields is valuable for understanding what “foundation inference” needs to generalize.
4. This manuscript is generally well organized, making it easy for readers to follow.

**Weaknesses:**
1. The pretraining distribution in the paper is significantly biased by construction. The pipeline explicitly discards divergent trajectories (magnitude >10^2), “focusing training on bounded regimes.” Although pragmatic, this may remove precisely the unstable regimes where robust system identification is hardest and most important, and it weakens claims of broad generalization from the prior.
2. The "local" functional representation is presented as an advantage, but it acts as a double-edged sword and still raises concerns even with the trade-off discussions. As demonstrated by the frictionless pendulum example, FIM-ODE overfits to the local region and utterly fails to preserve global geometric invariants such as closed conservative orbits.
3. The generalization narrative in this paper depends heavily on metric thresholds. On ODEBench generalization with the stringent threshold R^2>0.9, performance is low for both models and the head-to-head results are mixed rather than clearly favorable. The advantage becomes clearer mainly when relaxing the criterion to R^2>0.8. This seems shaky for a strong generalization claim.
4. Current scope is restricted to low-dimensional autonomous ODEs (≤3D), and the real-data experiment explicitly projects down and shows severe failure modes when information is lost (large errors for some subjects before finetuning). This limits immediate impact outside niche low-dim settings. Also, zero-shot performance in the low-data regime is described as sensitive to noise realizations, which constrains reliability in exactly the regimes where amortization is most attractive.

---

> ### Author Rebuttal · Authors · 2026-03-31
>
> We thank the reviewer for the careful and constructive comments. We address the main points below and will revise the paper accordingly.
>
> **W1:** We agree that our prior is biased by design. Trajectories that rapidly diverge not only create numerical-instability issues during training, but also correspond to regimes that are less representative of the bounded behaviors targeted here. Our aim is therefore not to model arbitrary ODEs uniformly, but to bias the prior toward low-dimensional polynomial systems with valid solutions that capture local patterns likely to recur in many non-polynomial systems. In that sense, the bias is deliberate and is one of the main design principles of the method.
>
> **W2:** We agree with the reviewer. The frictionless pendulum example is included precisely to show how and why FIM-ODE can *fail* on OOD systems. In that case, the context does not cover enough of the global structure of the vector field, so FIM-ODE fails to reconstruct the second trajectory. This is exactly the other side of the locality trade-off. In contrast, the CDIMA example shows that when the context covers the relevant region well, FIM-ODE can still yield a good approximation even out of distribution.
>
> **W3/Q5:** We agree that the current ODEBench discussion (Experiment 1) relies too strongly on the thresholded $R^2$ metric inherited from ODEFormer. We use this protocol for fairness, but we do not intend to base our claims on it alone. For completeness, we have now recomputed the relevant results using median MSE and MSE quantiles. For this, we separate the dimensions to account for different scales of the MSE. These new results follow, are consistent with Tables 1,2, and will be added to the revision.
>
>
> |Dim.|ODEF Rec (Med)|ODEF Rec (0.95q)|FIM Rec (Med)| FIM Rec (0.95q)|ODEF Gen (Med)| ODEF Gen (0.95q)|FIM Gen (Med)|FIM Gen (0.95q)|
> |---|:--:|:--:|:--:|:--:|:--:|:--:|:--:|:--:|
> |1D|0.0103|8.36|6.63e-03|3.29|0.216|445.71|0.289|859.37|
> |2D|0.0855|24.68|0.0214|2.22|0.836|566.72|0.551|40.88|
> |3D|13.79|1749.21|8.33|2492.35|15.18|2496.89|9.57|2575.50|
>
>
> **W4:** We agree that the current scope is limited and already acknowledge this in the paper. We disagree, however, that low-dimensional systems are scientifically marginal. Many important continuous-time phenomena admit effective low-dimensional descriptions, including epidemics, predator–prey dynamics, neuron excitability, chemical oscillations, pharmacokinetics, and reduced-order models, more broadly. Experiment 3 (MoCap) also illustrates one route by which low-dimensional ODEs can still be useful for high-dimensional observations.
>
> *Regarding noise sensitivity*, this is not specific to FIM-ODE. ODEFormer shows similar behavior. Our interpretation is that, in such regimes, the available context is not informative enough for robust zero-shot inference. To support this point, we repeated Experiment 2 while progressively increasing the amount of context by adding trajectories with random initial conditions, sampled from a normal distribution with variance 0.1 around the original VDP and FHN initial conditions. All trajectories share the same corruption mechanism and the same time grid.
>
> |#Traj.|VDP(T1)Med.|VDP(T1)0.95q|VDP(T2)Med.|VDP(T2)0.95q|FHNMed.|FHN0.95q|
> |---|---:|---:|---:|---:|---:|---:|
> |1|0.3792|1.9808|2.2443|4.9921|0.3263|1.1202|
> |2|0.3014|1.4856|1.4049|3.7391|0.2271|0.7691|
> |3|0.2617|1.4982|1.1946|3.3299|0.1931|0.5267|
> |9|0.3072|0.9138|1.0498|2.2369|0.1520|0.2417|
>
> These findings show that richer context substantially stabilizes inference, improving the Median MSE (Med) and reducing the 0.95-quantile of MSE.
>
> **Q1:** We agree that vector-field-level diagnostics are important. To address this, we now:
>
> (i) compute vector-field RMSE on grids defined around the original complete trajectory data by Hegde et al. (2022), for VDP and FHN in Experiment 2. The results follow:
>
> ||VDP1|VDP2|FHN|
> |-|-|-|-|
> |FIM-ODE|2.73|2.81|3.27|
> |ODEFormer|4.01|4.58|4.13|
>
> (ii) will also provide qualitative vector-field plots for all ODEBench systems, analogous to Figs. 2a and 2b.
>
> **Q2:** We did not study solver sensitivity systematically in the current version. In practice,
> 1. Euler is used only to generate the synthetic pretraining data, with a small step size.
> 2. All experiments in Section 5 solve the inferred ODEs with scipy.integrate.solve_ivp.
> 3. The target datasets are the original datasets from ODEBench and Hegde et al. (2022).
>
> We will clarify this in the main text.
>
> **Q3:** In practice, uncertainty weighting was essential for stable training: we were unable to train the model reliably without it. We also have direct evidence that richer context improves inference: Appendix E.5 already shows this on synthetic polynomial data, and we now add the corresponding multi-trajectory analysis for Experiment 2 (see our reply to W4).
>
> **Q4:** Not really. We thank the reviewer for this suggestion and we will investigate it in future work.

---

> > ### Author Rebuttal · Reviewer_7N6t · 2026-04-03
> >
> > Thank you to the authors for their efforts during the rebuttal stage. I tend to keep my original score due to concerns about the practical applicability of the proposed method.

---

> > > ### Author Response · Authors · 2026-04-06
> > >
> > > We would like to emphasize that the main contribution of this work is to establish *amortized ODE inference as a viable paradigm*.
> > >
> > > ODEFormer, an ICLR 2024 Spotlight paper, was the first model to amortize ODE inference. It showed zero-shot ODE inference on systems of **dimension at most 4**. That is, also in the *low-dimensional* regime. We took that work as motivation and proposed a strong alternative that can likewise perform zero-shot ODE inference, supports multi-trajectory conditioning, can be finetuned easily, and outperforms ODEFormer while being about ten times smaller and pretrained on about eighty times fewer systems.
> > >
> > > In other words, FIM-ODE addresses the same low-dimensional regime as prior zero-shot ODE inference work, but does so more efficiently, more flexibly, and with stronger empirical performance.
> > >
> > > *Why does this matter for practical applicability?*
> > >
> > > As we state in the introduction (very first paragraph), *amortized inference via neural networks* is rapidly changing AI and machine learning. ODE inference is a fundamental part of scientific modelling, and some of the most flexible existing methods available to date rely on neural variational inference or neural ODEs, both of which are well known to be challenging to train (see our related work section).
> > >
> > > Pretrained models for ODE inference offer an alternative to the repeated retraining of complex models. Their ease of finetuning also opens the door for non-ML experts to use state-of-the-art inference models.
> > >
> > > The natural next step is therefore to extend these methods to higher-dimensional systems, as we discuss in the limitations section, and as we plan to do in follow-up work.

---

### Official Review · Reviewer_A8gu · 2026-03-12

**Soundness:** 2
**Presentation:** 2
**Significance:** 2
**Originality:** 2
**Overall Recommendation:** 3
**Confidence:** 4

**Summary:**

This work introduces a new approach, FIM-ODE, as a foundation model of estimating vector fields from sparse trajectories of ODEs. The authors propose a prior distribution over ODE vector fields realizations of up to 3 dimensions using low-order polynomial expansions, and adapt an existing architecture to learn the vector fields in-context. The authors compare their approach to ODEformer, a contemporary method estimating ODE vector fields symbolically on benchmark tasks, as well as evaluating their approach on a human motion capture task.

**Compliance With Llm Reviewing Policy:**

Affirmed.

**Final Justification:**

Following the rebuttal from the authors, which included additional empirical evaluations showing good performance of their method when several trajectories are provided as context, I have higher confidence that the approach of the authors works, and have raised my score from 2 to 3. My main concern remains the scalability to higher dimensional differential equations.

**Key Questions For Authors:**

1. Could the authors provide the runtimes of the model (pretraining/finetuning/inference of FIM-ODE, as well as baselines?)
2. An advantage of the approach is that the trained model can be conditioned on several trajectories from the ODE. However, as far as I understand from the tasks presented in this work, the model is always conditioned on one trajectory. How would conditioning on several trajectories improve the pretrained model (i.e. without finetuning) on the harder tasks of Van der Pol and FHN?
3. The authors train an auxillary prediction head U, which the authors state is “interpreted as the log-variance”. By this, do the authors mean that this is the log-variance of the prediction in the vector field? If so, can this uncertainty also be used outside of training, e.g. to predict uncertainty in the model’s prediction of the trajectory?
4. I am not entirely clear as to how inference is performed. Given some trajectory and an initial point $x(0)$, the model outputs a vector $f_\theta(x(0))$, and since the authors consider Euler integrations, the system is evolved with $x(\Delta t) = x(0) + \Delta(t) f_\theta(x(0))$. Is the new transition from $x(0)$ to $x(\Delta t)$ now also added to the context before computing the next vector field, or is the context kept fixed? The former has the advantage of potentially improving the consistency of individual trajectories, but comes at additional computational cost as the context needs to be encoded again for each timestep. Could the authors elaborate on this point?
5. I am not sure why the authors only consider Euler integration of ODEs, especially when they consider stiff ODEs such as Van der Pol and FHN. Is there a computational constraint that I am missing?

**Limitations:**

Yes.

**Strengths And Weaknesses:**

## Strengths:

1. One of the authors’ key insights is to learn the local vector field as opposed to the global vector field, which improves the generalization abilities of their models to new trajectories. This allows the author to define a formulation for the task of learning the vector fields of Ordinary Differential Equations which is novel and interesting.
2. The approach of the author outperforms ODEformer, the main baseline, across the majority of tasks.
3. The authors additionally present an approach for finetuning their approach on specific trajectories which significantly improves performance in the sparse data regime.

## Weaknesses:

1. As noted by the authors, the restriction of FIM-ODE to 3-dimensional systems is very limiting, meaning that the model can realistically only be applied to a very limited set of mostly toy problems.
2. I am surprised that for tasks involving oscillators, particularly VdP and FHN, FIM fails the task completely without finetuning. This is because the provided context (the first few seconds of the trajectory) already contains one full oscillation from these oscillating systems, so the task becomes to repeat this oscillation, which should be learnable from the context alone without finetuning. I am also somewhat concerned about the inclusion of results with “selected noise realizations” in Table 4, as opposed to simply reporting mean and standard deviations in the error across runs/trajectories.
3. There appears to be some mismatch between the framing of the method and how it is applied in the experiments section. It is not possible to generalize from seeing one trajectory of an ODE to being able to predict the trajectory at other locations. when predicting vector fields away from the observations, whether the predictions are correct or not mostly depends on whether the inductive bias in the training data of the model matches the specific ground truth. This is demonstrated well in Figure 2a, where observing one limit cycle of the ODE does not tell us what happens away from this limit cycle, so the fact that in this case FIM-ODE guesses “wrong” is not surprising. It is perhaps more reasonable to condition on several distinct trajectories at inference time, which the authors do not seem to investigate (see questions). Instead, most of the experiments focus on reproducing one trajectory from partial observations of the first few timesteps. However, this task does not strongly benefit from having a foundation model that is finetuned per trajectory.

---

> ### Author Rebuttal · Authors · 2026-03-31
>
> We thank the reviewer for the helpful comments. We clarify below the intended scope of the paper and the interpretation of the experiments.
>
> **W1:** From the abstract onward, we state that our goal is zero-shot inference of **low-dimensional** ODEs. We do not claim general high-dimensional ODE inference, and we explicitly acknowledge this as a limitation. We disagree, however, that low-dimensional systems are mostly toy problems. Many important continuous-time phenomena admit effective low-dimensional descriptions, including population growth, epidemics, neuron excitability, chemical oscillations, pharmacokinetics, etc.
>
> **W2/Q2:** Let us first note that neither FIM-ODE nor ODEFormer solves Experiment 2 satisfactorily in zero-shot mode. Our interpretation is that the available context does not contain enough information to estimate the vector field reliably beyond the observed region. This may be due to the noise level, the mismatch between evaluation and pretraining corruption, and the fact that this low-data regime is itself unseen during pretraining. To support this point, we repeated Experiment 2 while progressively increasing the amount of context by adding trajectories with random initial conditions sampled from a normal distribution with variance \(0.1\) around the original VDP and FHN initial conditions. All trajectories share the same corruption mechanism and time grid.
>
> |#Traj.|VDP(T1)Med.|VDP(T1)0.95q|VDP(T2)Med.|VDP(T2)0.95q|FHNMed.|FHN0.95q|
> |---|---:|---:|---:|---:|---:|---:|
> |1|0.3792|1.9808|2.2443|4.9921|0.3263|1.1202|
> |2|0.3014|1.4856|1.4049|3.7391|0.2271|0.7691|
> |3|0.2617|1.4982|1.1946|3.3299|0.1931|0.5267|
> |9|0.3072|0.9138|1.0498|2.2369|0.1520|0.2417|
>
> These results show that richer context substantially stabilizes inference by improving the median MSE (Med) and reducing high-error cases (0.95-quantile of MSE). Regarding the concern about selected noise realizations, we did report means and standard deviations in Appendix Table 7 and refer to them in the main text.
>
> **W3:** We agree with the reviewer’s conceptual point. The trajectory-generalization task in Experiment 1 was introduced in ODEFormer, and we follow it for fairness. Performance on that task necessarily depends on whether the context constrains the vector field along the second trajectory. Thus, the behavior in Fig. 2a is not surprising. Similarly, Experiment 2 also evaluates trajectory reconstruction and therefore shares the same limitation.
>
> To address this more directly, we now:
> (i) repeat Experiment 2 with additional context trajectories, showing that richer coverage stabilizes inference (W2); and
> (ii) compute vector-field RMSE against ground-truth for both FIM-ODE and ODEFormer, on a grid around the original trajectory data of Hegde et al. (2022):
>
> ||VDP1|VDP2|FHN|
> |-|-:|-:|-:|
> |FIM-ODE|2.73|2.81|3.27|
> |ODEFormer|4.01|4.58|4.13|
>
> FIM-ODE thus outperforms ODEFormer in vector field inference. These analyses better reflect the system-identification capabilities of FIM-ODE.
>
> **Q1:** We already provide part of the runtime information. Pretraining took five days on four NVIDIA A40 GPUs (48 GB each), and finetuning typically used 200 or 800 epochs, although in many cases as few as 20 already gave strong results. We agree, however, that the paper does not state finetuning hardware, wall-clock finetuning times, or inference times. We therefore summarize them here: finetuning used a single NVIDIA A100 (40 GB, about 3 GB used per task); FHN finetuning took \(<1\)h, MoCap 15--30 min, and VDP \(<1\)h20min; inference on an M4 MacBook Air (16 GB) takes about 16 ms to predict a vector field at 512 locations from one context trajectory with 200 observations. We will add this to the appendix.
>
> **Q3:** The auxiliary head \(U\) is essential during training because it compensates for scale mismatch across regions of a vector field. Empirically, however, we did not find convincing evidence that it yields calibrated predictive uncertainty in low-data regimes. During training, the overall uncertainty level decreases, while its spatial pattern remains broadly similar. We will add visualizations in the appendix.
>
> **Q4:** In our current setup the context is kept fixed. Given \(K\) trajectories, FIM-ODE produces a continuous neural representation of the vector field, which can then be queried at arbitrary \(x\). We do **not** iteratively add generated transitions back into the context. The reviewer’s proposed iterative update scheme is interesting and may improve forecasting, but would require recomputing the context representation repeatedly.
>
> **Q5:** This confusion is caused by our presentation. In our implementation, Euler is used only to generate the synthetic pretraining data, with a very small integration step. All experiments in Section 5 solve the inferred ODEs using `scipy.integrate.solve_ivp` (default explicit Runge--Kutta of order 5). We will make this distinction explicit in the revised paper.

---

> > ### Author Rebuttal · Reviewer_A8gu · 2026-04-03
> >
> > I thank the authors for clarifying my questions and for the additional empirical evaluations. I particularly appreciate the additional evaluations on several trajectories, which I believe are a more natural evaluation of these types of methods, as well as evaluations of the RMSE of the vector field. These convince me that the method can indeed work empirically in learning vector fields, which the 1-trajectory evaluations presented before did not. I will increase my score accordingly. However, the main limitation of this work, as indeed pointed out by the other reviewers, is the limited dimensionality of the systems considered. I concede the author's point that many real-world systems have **effective** low dimensionalities, but then the responsibility of the authors would be to show how to couple learning the low-dimensional latent dynamics with learning what the effective latent dimensions of the system are. As pointed out by the other reviewers, the motion capture example presented in this work shows exactly the failure mode of using a "naive" projection (in this case, PCA), that is decoupled from FIM-ODE. Even with relatively low amounts of information loss, the trajectories cannot be recovered. Without presenting a robust approach for learning the low-dimensional dynamics of high-dimensional systems, the claim that many systems have low effective dimensionalities is not sufficient for me to recommend acceptance of this iteration of this work.

---

> > > ### Author Response · Authors · 2026-04-06
> > >
> > > We thank the reviewer for their careful follow-up and for acknowledging the additional empirical evaluations. We nonetheless believe that (i) dynamical system inference (given a coordinate system) and (ii) coarse-graining (or coordinate inference) should be studied as *distinct problems*. We understand that neural networks have enabled end-to-end approaches to these tasks. However, such approaches require learning two transformations simultaneously: the transition function that captures the dynamics, and the encoding-decoding maps. When trained jointly, these components become coupled, which can lead to complex and unstable training procedures. We will expand on these issues in a dedicated related work section.
> > >
> > > We view the amortization of dynamical system inference precisely as a way to *decouple* these two problems. Indeed, recent work has pointed out that this may be a promising route to accelerate coarse-graining: namely, to leverage the inductive biases encoded in the pretrained weights of foundation models, while learning only autoencoders on top of fixed-weight FIM-like models (Hinz et al., 2025).
> > >
> > > We would also like to remark that FIM-ODE was inspired by ODEFormer, which, to our knowledge, was the first model capable of zero-shot ODE inference, but was likewise **evaluated on systems of dimension at most 4**. FIM-ODE provides a strong alternative: it can also perform zero-shot ODE inference, supports multi-trajectory conditioning, can be finetuned easily, and outperforms ODEFormer while being approximately ten times smaller and pretrained on about eighty times fewer systems.
> > >
> > > FIM-ODE therefore addresses the same low-dimensional regime as prior zero-shot ODE inference work, but does so more efficiently, more flexibly, and with stronger empirical performance.
> > >
> > > **References**
> > >
> > > - Hinz et al. 2025: Towards Fast Coarse-graining and Equation Discovery with Foundation Inference Model

---

### Official Review · Reviewer_dzSS · 2026-03-13

**Soundness:** 3
**Presentation:** 3
**Significance:** 3
**Originality:** 3
**Overall Recommendation:** 5
**Confidence:** 3

**Summary:**

This paper introduces FIM-ODE, a pretrained Foundation Inference Model (FIM) for inferring vector fields of ordinary differential equations (ODEs) from noisy, sparse trajectory data.
Their model is a transformer-based neural operator architecture (8M parameters) that outputs continuous, locally-evaluated vector field estimates.
The model is pretrained on a prior distribution over polynomial ODEs of degree ≤ 3 in dimensions 1–3.
The primary baseline FIM-ODE that compares against is ODEFormer (86M parameters).
The paper discusses the merits and tradeoffs between global representation (ODEFormer's symbolic regression) and local representation (FIM-ODE vector field estimation), which is local because it aims to approximate the vector field in regions of state space visited by the trajectories observed in the data.
They discuss two tasks:

- trajectory reconstruction, measuring how accurately the inferred vector field reproduces a clean reference trajectory from the ground truth initial conditions that generated the noisy data in context
- trajectory generalization, measuring how accurately the inferred vector field produce trajectories from initial conditions not present in context.

Experiment 1 assesses these two tasks on 61 dynamical systems from the ODEBench suite.
The experiment results show FIM-ODE consistently outperforming ODEFormer on trajectory reconstruction, and comparable at trajectory generalization.
Paper notes that ODEBench includes non-polynomial ODE systems, which are out-of-distribution for the FIM-ODE the degree-3 polynomial prior, suggesting the performance on the ODEBench experiments indicate OOD generalization (to non-polynomial systems).

Experiment 2 assesses OOD generalization to context lengths shorter than seen during training.
They test this on two systems (Van der Pol and FitzHugh Nagumo).
In addition to the ODEFormer baseline, they also compare to multiple state-of-the-art baselines trained on the per-dataset paradigm (including variants of Neural ODEs, SDEs, and GP-based approaches).
In this zero-shot setting the amortized methods (ODEFormer and FIM-ODE) perform worse than the per-dataset baselines, with FIM outperforming ODEFormer on one system (FHN) but not the other (VDP-uniform observations).
Further, they fine tune the FIM-ODE on this data and the FIM-ODE-finetuned outperforms the zero-shot models and performs comparably to the per-dataset methods.

Experiment 3 considers application to real-world trajectories of human motion capture.
Since this data exceeds FIM-ODE's 3-dimension maximum, the trajectories are projected to 5 dimensions via PCA, and the first 3 principal component dimensions are used.
The results report the MSE across 3 subjects for short and long horizon trajectory prediction, comparing zero-shot and fine-tuned ODE-FIM against the per-dataset baselines in experiment 2.
The zero-shot results are mixed, as ODE-FIM performs comparable to the baselines on 1 subject, but significantly worse on two.
After fine-tuning ODE-FIN performs comparable to the baselines across all subjects.

**Compliance With Llm Reviewing Policy:**

Affirmed.

**Final Justification:**

I want to thank the authors for their thorough and productive consideration of my concerns. Their rebuttals and replies have addressed my concerns.

**Q1:** Their reply acknowledging the relatively weak 3D performance is convincing, that a significant portion of the 3D systems are chaotic. This will be visible in the promised per-system results table. I also read the reply to Reviewer 7N6t, the inclusion of scores wrt MSE metric strengthens the experimental results.

**Q2:** The authors confirm that they were referencing the published ICLR paper, not the updated arXiv version, and that the conference paper omitted the reproducibility details that I was concerned about. This is unfortunate, but the noise range chosen in their experiments are still within the ODEFormer training distribution. I'd consider their proposed additional ablation with $\sigma = 0.1$, OOD for FIM-ODE but ID for ODEFormer, to be a nice-to-have but not critical for acceptance.

**Q3:** The ablations discussed in this questions, which authors acknowledge and will provide, will strengthen the paper's discussion by disentangling the different sources of distributional mismatch.


I also want to address my W7 weakness that I did not explicitly reply to previously. I included the low-dimensionality as a practical limitation in my review for completeness. The authors explicitly acknowledge this limitation in the paper, identified as future research direction, include the MoCap experiment to demonstrate potential utility even in higher-dimensional problems, and their primary baseline (ODEFormer) is also limited to low-dimensional systems. I believe this work contributes to the field and improves upon the established (similarly limited) baseline.

Given the authors have more than sufficiently addressed my concerns, I am changing my overall recommendation to Accept.

**Key Questions For Authors:**

1. The ODEBench experiments 5.1 suggest that FIM-ODE is able to generalize to the OOD systems in the benchmark (non-polynomial, degree>3). The paper claims that Tables 1 & 2 results support this OOD generalization. However, the tables report aggregate success rates over all ~61 systems without distinguishing polynomial (degree <=3) from polynomial (degree >3, OOD) and non-polynomial (OOD) subsets. Could the authors report Tables 1 & 2 broken down by system type (in-distribution vs OOD-polynomial vs OOD-nonpolynomial)? Without this breakdown, it is impossible to determine whether the aggregate numbers reflect the claimed OOD generalization or a strong in-distribution performance obscuring an OOD failure.


2. The Gaussian noise corruption procedure described in ODEFormer (d'Ascoli et al., 2024, Section 3) defines its noise as distributed $\mathcal{N}(0, \sigma)$ with "noise-level" $\sigma$ sampled uniformly from `[0., 0.1]`.
(It's possible this is just an unfortunate notation/typo in ODEBench, and that their $\sigma$ also refers to standard deviation.)
The data corruption procedure described in Appendix B in defines the noise as Gaussian distributed $\mathcal{N}(0, \sigma^2)$ with $\sigma$ sampled uniformly `[0, 0.06]`.
It appears the Gaussian noise level is reduced compared to the corrupting procedure in ODEBench. Can the authors clarify if this is intentional, and if so why the corruption procedure isn't identical to the ODEFormer training data? Do the authors believe the reduced noise training/evaluation of FIM-ODE compromises the direct comparison to ODEFormer?
Also, the caption in Figure 2 indicates $\sigma = 0.3$, was this intentionally larger noise than the training/evaluation distributions, and if so does it accurately reflect the reconstruction/generalization phenomenon in the evaluation?


3. This question is similar to the preceding question but relates to the noise corruption in the VDP and FHN experiments. The paper claims that these experiments are meant to evaluate low-data OOD. However, the noise-corruption procedure described here also appears to be OOD from the training distribution. For VDP the observations are "corrupted by additive Gaussian noise with variance $\sigma^2 = 0.05$". For FHN the observations are "corrupted by additive noise with variance $\sigma^2 = 0.025$". The FIM-ODE and ODEFormer training data is corrupted with multiplicative Gaussian noise, not additive.
Further, the noise levels selected correspond to $\sigma \approx 0.07$ " and  $\sigma \approx 0.16$ " respectively, which exceed the $0.06$ maximum noise-level in FIM-ODE's training data. Can the authors indicate if this is intentional?
If my understanding is correct, then the experimental results in this section are OOD in both low-data (discussed and intentional) and in the noising distribution (not mentioned), which complicates the interpretability of the results as only assessing the low-data performance. Further, the discussion describes sensitivity to noise seed, (e.g. FIM-ODE (Selected noise) is reported in Table 4, and reporting the best-performing noise sample in this way raises suspicion). Do the authors think it is possible that this sensitivity to noise realization is not entirely due to low-context, but because the noise corruption is OOD (either due to additivity, or unseen noise-levels)?

**Limitations:**

yes

**Strengths And Weaknesses:**

### Strengths

#### Soundness

- The paper clearly acknowledges limitations such as the dimensionality ceiling at $d=3$, the non-stationarity of the polynomial prior, and the highly variable zero-shot performance in Experiment 2.
- The uncertainty-weighted loss (Section 4.2) is appropriate and precedented with citation to prior work. It corresponds to a standard Laplace likelihood with a learned log-variance head, and motivation is explained, down-weighting uncertain regions while preventing degenerate solutions.
- Experiment 1 follows the established ODEBench protocol, and the evaluation methodologically is sound.


#### Presentation

- The paper is structured and written well. The problem statement and notation in (Section 3) is precise and easy to follow. The distinction between trajectory reconstruction and trajectory generalization is clearly maintained in the main text.
- The qualitative presentation in Figure 2 illustrates reconstruction and generalization phenomenon extremely well. Especially 2a, which illustrates when the proposed FIM-ODE fails to generalize to unseen initial conditions. This is an excellent presentation of the limitations of the polynomial prior. Illustrating and discussing the mechanism for this informative failure mode in the method is highly appreciated.
- Method, architecture, data generation procedure, and fine-tuning protocol are all described with sufficient detail for reproduction and assessment of experimental validity.
- The paper correctly positions itself within the FIM literature, amortized symbolic regression literature, and neural operator methods. The distinction between amortized and per-dataset inference paradigms is clearly articulated.

#### Significance

- Zero-shot ODE identification from sparse, noisy trajectories is broadly applicable and an interesting research direction. A successful pretrained model that can provide zero-shot estimates or used for finetuning has practical utility in settings where per-dataset optimization is expensive, or as a baseline for future research in this area.
- Achieving competitive/superior performance to ODEFormer at lower parameter count increases its utility for practitioners with constrained computational resources (for inference or fine-tuning), and could influence design choices for future amortized dynamical systems models.
- Their proposed polynomial-prior for pretrained model as a general initialization for downstream dynamical system tasks is a simple, this may inspire future work either using this prior directly as a simple foundation (e.g. for fine-tuning) or as a baseline for evaluating alternative more expressive priors (a stated future direction).

#### Originality

- Framing the pretraining distribution as a non-stationary GP over the space of polynomial vector fields provides a clean statistical interpretation for why the model generalizes.

### Weaknesses

#### Soundness

- The OOD generalization claim to non-polynomial systems in the ODEBench is not verifiable from the reported numbers in Tables 1 and 2. It's possible that strong performance on the in-distribution systems is obscuring consistent failure on the non-polynomial systems in the benchmark suite.
- The noise corruption convention apparently differs between FIM-ODE and ODEFormer, compromising their direct comparison.
- Noise magnitude and type mismatch between pretraining (multiplicative noise with $\sigma \leq 0.06$) and zero-shot evaluation Experiment 2 (additive noise with $\sigma \leq 0.07$ or $0.16$). This is not discussed in the text and compromises the ability to assess the stated interpretation of the zero-shot evaluations (as caused by OOD low-context).
- The claim that global vs local representation is primarily responsible for the differences in results is confounded by uncontrolled differences. In particular, FIM-ODE and ODEFormer differ in model size, training data size, and training distribution. An ablation that controls these aspects would allow more direct assessment of the claims around global vs local merits.



#### Presentation

- The distinction between "trajectory generalization" (i.e. to new initial conditions) and the various "OOD generalization" (context length, non-polynomial systems) is confusing and sometimes not clearly distinguished.  This ambiguity makes it easy to read Table 2 as evidence for OOD transfer, discussed in the paragraph beginning at line 298. Consistently qualifying uses of "generalization" would improve the presentation.
- Figure 2 caption states $\sigma = 0.3$, which exceeds the pretraining range (max 0.06) and evaluation range (max 0.05). This is possibly a typo for 0.03, but if not this complicates the utility of Figure 2 as being indicative/explanatory for the results reported in Tables 1 and 2, or the discussion of global vs local representations.


#### Significance

- Dimensionality ceiling of $d\leq 3$ limits the impact for ODE-FIM to be applied. The dimensionality reduction via PCA in the MoCAP experiments partially address this, with some negative results included, but in general dimensionality reduction may not be appropriate for higher-dimensional dynamical systems.
- The 100 seed statistics in Appendix E.2 Table 7 reveals substantial variance across random seeds. This sensitivity to noise in the zero-shot performance limits the application for practitioners who are unable to fine-tune.


#### Originality

- The architectural and training ideas are closely related to FIM-SDE. This is explicitly acknowledged in the text.

---

> ### Author Rebuttal · Authors · 2026-03-31
>
> We thank the reviewer for the careful reading and constructive comments.
>
> **W1/Q1:** The reviewer is right that Tables 1, 2 report only aggregated metrics. To make the OOD claim explicit, we now split ODEBench into systems with polynomial vector fields of degree at most $3$ (ID) and the rest (OOD), and recompute the metrics separately.
>
> **Recon. \% ($R^2>0.9$)**
> |Sys.|FIM $(0,0)$|FIM $(0.5,0.05)$|ODEF $(0,0)$|ODEF $(0.5,0.05)$|
> |-|-:|-:|-:|-:|
> |1D ID (11)|100.0|86.4|86.4|77.3|
> |1D OOD (12)|100.0|95.8|91.7|87.5|
> |2D ID (15)|93.3|76.7|50.0|46.7|
> |2D OOD (13)|92.3|76.9|69.2|73.1|
> |3D ID (9)|16.7|5.6|11.1|11.1|
> |3D OOD (1)|100.0|100.0|100.0|100.0|
>
> **Gen. \% ($R^2>0.9$)**
> |Sys.|FIM $(0,0)$|FIM $(0.5,0.05)$|ODEF $(0,0)$|ODEF $(0.5,0.05)$|
> |-|-:|-:|-:|-:|
> |1D ID (11)|54.5|54.5|59.1|54.5|
> |1D OOD (12)|50.0|54.2|37.5|41.7|
> |2D ID (15)|26.7|23.3|20.0|20.0|
> |2D OOD (13)|15.4|11.5|19.2|15.4|
> |3D ID (9)|0.0|0.0|5.6|5.6|
> |3D OOD (1)|0.0|0.0|0.0|0.0|
>
> None of these systems are OOD for ODEFormer. These tables make two points explicit. First, for reconstruction, FIM-ODE is strong on both ID and OOD systems, in 1D--2D, and consistently outperforms ODEFormer. *Hence the aggregate gains are not driven only by ID systems*. Second, for generalization, the picture is more mixed, as expected: in 1D FIM-ODE is similar on ID/OOD, while in 2D OOD performance weakens, consistent with the need to extrapolate beyond the region constrained by the context.
>
> **W2/Q2:** In lines 191--200 we state that we use multiplicative Gaussian noise and random subsampling, $y_i=(1+\epsilon)x_i, \epsilon\sim\mathcal N(0,\sigma^2),$ with Bernoulli masking probability $\rho\in[0,0.5]$. Appendix B specifies $\sigma\in[0,0.06]$, which we will move to the main text. To the best of our knowledge, this matches ODEFormer's mechanism, although we could not locate their exact pretraining ranges for $\sigma$ and $\rho$. Our choice $\rho\in[0,0.5]$, $\sigma\in[0,0.06]$ was made to cover all target corruption settings.
>
> **W3/Q3:** We reproduce the setup of Hegde et al. (2022), which uses additive noise, for fairness. We agree that this introduces an additional mismatch relative to our pretraining distribution, and we will state this explicitly. Thus, Experiment 2 is OOD not only because of the low-context regime, but also because of the corruption mechanism.
>
> **W4:** We agree that FIM-ODE and ODEFormer differ in more than local vs. global representation: they also differ in architecture, objective, model size, training data size, and training distribution. A full ablation would require a separate study and is beyond the scope of this paper. Our goal in Experiment 1 was narrower: to illustrate the regimes in which local and global representations are respectively favorable.
>
> **W5:** We will clarify the distinction between (i) generalization to new initial conditions and (ii) OOD generalization relative to the pretraining distribution.
>
> **W6:** The caption of Fig. 2 contains a typo: it should read $\sigma=0.03$.
>
> **W7:** From the abstract onward, we state that our goal is zero-shot inference of low-dimensional ODEs, not general high-dimensional ODE inference. At the same time, **low-dimensional systems are far from toy settings**: many important phenomena admit effective descriptions in dimensions up to three, including epidemic spread, predator--prey dynamics, neuron excitability, chemical oscillations, pharmacokinetics, and many reduced-order models. MoCap also illustrates one route by which low-dimensional ODEs can still be useful for high-dimensional data.
>
> **W8:** This issue is not specific to FIM-ODE; ODEFormer shows similar variance. Our interpretation is that, in this regime, the available context is not informative enough for either model to *zero-shot estimate* the vector field reliably away from the observed region. To support this point, we repeated Experiment 2 while progressively increasing the amount of context by adding trajectories with random initial conditions
>
> |#Traj.|VDP(T1)Med.|VDP(T1)0.95q|VDP(T2)Med.|VDP(T2)0.95q|FHNMed.|FHN0.95q|
> |---|---:|---:|---:|---:|---:|---:|
> |1|0.3792|1.9808|2.2443|4.9921|0.3263|1.1202|
> |2|0.3014|1.4856|1.4049|3.7391|0.2271|0.7691|
> |3|0.2617|1.4982|1.1946|3.3299|0.1931|0.5267|
> |9|0.3072|0.9138|1.0498|2.2369|0.1520|0.2417|
>
> These results show that richer context stabilizes inference by improving the median MSE and reducing high-error cases.
>
> **W9:** In line 230 we state that the architecture is adapted from FIM-SDE, but the ODE inference problem is substantially different from the SDE one. In SDEs, stochastic forcing can probe much more of the state space and leads to stronger theoretical identifiability. This is absent in ODEs. As a result, constructing a useful prior over ODEs is considerably more delicate. We therefore view this work not as a straightforward transfer of FIM-SDE, but as a nontrivial extension of the FIM framework to a setting with different identifiability and prior-design challenges.

---

> > ### Author Rebuttal · Reviewer_dzSS · 2026-04-03
> >
> > **Q1:** Thank you for providing the the split Tables 1 and 2. This resolves my concern about ability to assess OOD claims. In addition to splitting IID/OOD, you also split by system dimension. This revealed some interesting results not visible in the original aggregated tables.
> > Consider discussing the following observations in your revision:
> > - 3D OOD results on both FIM and ODEF achieve 100% Recon. but 0% Gen. I understand there is only 1 system in this 3D OOD case.
> > - 3D IID, for Recon.%  FIM(0,0) performs better than FIM(0.5,0.05), which is expected. However, ODEF (0,0) vs (0.5,0.05) shows no change. Similarly, for Gen.% the FIM and ODEF remain unchanged between the 0. and 0.5 settings.
> > - Performance weakens as dimensionality increases for both FIM and ODEF. Both models performance in 3D is quite weak compared to their 1D/2D performance. However, the dropoff from 1D to 2D is more pronounced in ODEFormer than FIM.
> >
> > Consider providing a full breakdown of these results, i.e. with no aggregation over systems, in the appendix. This would allow assessing which specific systems are performing poorly.
> >
> > Do you think the poor performance on the 3D systems weakens your claims that FIM handles 3D systems?
> >
> > **Q2:** The ranges I referenced in my question, $\sigma = [0, 0.1]$ and $\rho= [0., 0.5]$ are in their "Corrupting data" paragraph of Section 3. So your $\rho$ matches, but FIM was trained with lower noise levels $\sigma$ than ODEFormer. I believe this weakens the direct comparison, e.g. of parameter count, given that ODEFormer was trained on a harder task due to increased noise level. Can you discuss what impact the lower noise level vs ODEFormer is expected to have when interpreting their relative performance / parameter size?
> >
> > **Q3:** Yes, acknowledging the OOD due to low-context AND corruption mechanism in Experiment 2 partially addresses the issue. However, it remains unclear whether the results are primarily explained by the corruption-OOD vs the context-OOD. Ablating over both OOD separately would be illuminating.
> >
> > Thank you for addressing my weaknesses as well.

---

> > > ### Author Response · Authors · 2026-04-06
> > >
> > > We thank the reviewer for engaging in discussion with us.
> > >
> > > **@Q1**: As suggested by the reviewer, we will provide a full breakdown of these results in the appendix. We would also like to draw the reviewer’s attention to our reply to **W3/Q5** of **Reviewer 7N6t**, where we report the scores of Experiment 1 with respect to the MSE metric. These results are consistent with, and complementary to, Tables 1 and 2, and they show that FIM-ODE outperforms ODEFormer *across all dimensionalities* in both trajectory reconstruction and trajectory generalization.
> > > We will also include the MSE scores with the ID vs. OOD system split in the appendix, and we will expand on their interpretation.
> > >
> > > The weaker performance on 3D systems is due to their greater complexity. Four out of the eleven target systems are chaotic, and are therefore naturally more sensitive to noise and to errors in vector field estimation.
> > >
> > > **@Q2**: We have now identified the detail the reviewer was referring to. Indeed, in the arXiv version of ODEFormer, the authors state that they used $\sigma \in [0, 0.1]$. This detail unfortunately does not appear in the ICLR version.
> > >
> > > Although it is difficult to infer precisely what effect this mismatch has on model performance — just as it is difficult to isolate the precise effect of our different priors — we can at least say that both models are compared on systems for which the noise corruption is ID (Experiment 1) and OOD (Experiment 2).
> > > We propose to include an additional experiment in Tables 1 and 2 in which the noise corruption is fixed to $\sigma = 0.1$ (that is, ID for ODEFormer and OOD for FIM-ODE). We will again provide these results in the appendix.
> > >
> > > **@Q3**: We agree with the reviewer and will include such an ablation in the appendix of the paper.

---

### Decision · Program_Chairs · 2026-04-30

**Decision:**

Accept (regular)

**Comment:**

This paper introduces FIM-ODE, a pretrained model for inferring ODE vector fields from noisy and sparse trajectories. The paper focuses on amortized inference using a local vector field representation rather than a global symbolic one. Overall, an important concept outlined by the article is the tradeoff between local and global representations for system identification. The method is clearly described and technically sound, and the evaluation on ODEBench seems reasonable. I also find synthetic data generation and the simple polynomial prior appealing, as they provide a clean, intuitive foundation for pretraining while still supporting strong empirical performance. Reviewers generally agreed that the model is well motivated and that the empirical results are solid, especially for trajectory reconstruction, where FIM-ODE consistently outperforms ODEFormer while using far fewer parameters and a simpler pretraining setup. The finetuning results are also a strength, showing that the pretrained model provides a useful initialization in more challenging settings.

The main concerns relate to scope and interpretation rather than correctness. The restriction to low-dimensional systems limits applicability, and the zero-shot results in low-data settings can be unstable. There were also questions about how to interpret the OOD generalization claims and the role of differences in noise and training distributions. The authors addressed several of these issues in the rebuttal by providing additional clarifications. Some broader concerns about positioning and comparisons to alternative approaches remain and should be addressed with care, but overall, the paper makes a solid contribution within the setting it targets and improves on prior amortized baselines in a meaningful way.